# Enforcing Latent Euclidean Geometry in Single-Cell VAEs for Manifold Interpolation

Alessandro Palma [* 1 2]  Sergei Rybakov [* 1 2 3]  Leon Hetzel [* 1 2 4]  Stephan Günnemann [2 4]  Fabian J. Theis [1 2 4 5]

## Abstract

Latent space interpolations are a powerful tool for navigating deep generative models in applied settings. An example is single-cell RNA sequencing, where existing methods model cellular state transitions as latent space interpolations with variational autoencoders, often assuming linear shifts and Euclidean geometry. However, unless explicitly enforced, linear interpolations in the latent space may not correspond to geodesic paths on the data manifold, limiting methods that assume Euclidean geometry in the data representations. We introduce FlatVI, a novel training framework that regularises the latent manifold of discrete-likelihood variational autoencoders towards Euclidean geometry, specifically tailored for modelling single-cell count data. By encouraging straight lines in the latent space to approximate geodesic interpolations on the decoded single-cell manifold, FlatVI enhances compatibility with downstream approaches that assume Euclidean latent geometry. Experiments on synthetic data support the theoretical soundness of our approach, while applications to time-resolved single-cell RNA sequencing data demonstrate improved trajectory reconstruction and manifold interpolation.

## 1. Introduction

Generative models for representation learning, such as Variational Autoencoders (VAEs), have influenced computational sciences across multiple fields (Zhong & Meidani,

2023; Lopez et al., 2018; Griffiths & Hernández-Lobato, 2020). One reason is that real-world experimental data often poses significant modelling challenges as it is inherently noisy, high-dimensional, and complex (Sarker, 2021). As a solution, learning a compressed and dense latent representation of the data has gained traction in applied machine learning. For example, interpolations in well-behaved, low-dimensional data embeddings are useful for modelling the dynamics of complex systems, allowing insights into sample evolution over time (Džeroski & Todorovski, 2003; Bunne et al., 2022).

In particular, VAEs have shown great promise in representing both continuous and discrete data, as the decoder parameterises a flexible likelihood model. This flexibility has demonstrated unprecedented potential in cellular data (Lopez et al., 2018), particularly in gene expression, which is measured in counts that reflect the number of RNA molecules produced by individual genes and is collected through single-cell RNA sequencing (scRNA-seq) (Haque et al., 2017). Such a technique allows the measurement of thousands of genes in parallel, and the resulting vector describes the state of a cell across diverse biological settings (Regev et al., 2017). Leveraging latent interpolations within single-cell VAEs can provide insights into cellular state transitions, capturing dynamic changes in gene expression that reveal underlying biological processes.

The representation learnt by VAEs is tightly connected to Riemannian geometry, as one can see the latent space as a parametrisation of a low-dimensional manifold (Arvanitidis et al., 2021). When modelling latent cellular dynamics on single-cell data, popular approaches still rely on assuming Euclidean geometry in the representation space, for example by modelling cellular transitions through linear-cost Optimal Transport (OT) (Peyré et al., 2019; Klein et al., 2025; Tong et al., 2020). However, *building linear latent trajectories using the Euclidean assumption is sub-optimal when the data lies on a non-Euclidean manifold*, as straight latent lines do not necessarily reflect geodesic paths on the manifold induced by the decoder.

To learn effective interpolations on a single-cell manifold, we establish the following desiderata: (i) Approximate trajectories on intractable data manifolds via interpolations

[*]Equal contribution  [1]Institute of Computational Biology, Helmholtz Munich, Munich, Germany [2]School of Computation Information and Technology, Technical University of Munich, Germany [3]Lamin Labs [4]Munich Data Science Institute, Technical University of Munich, Germany [5]TUM School of Life Sciences Weihenstephan, Technical University of Munich, Germany. Correspondence to: Fabian J. Theis <fabian.theis@helmholtz-munich.de>.

*Proceedings of the 42nd International Conference on Machine Learning*, Vancouver, Canada. PMLR 267, 2025. Copyright 2025 by the author(s).

*Figure 1.* Visual conceptualisation of the FlatVI approach. The decoder of a single-cell VAE maps to the parameter space of a negative binomial statistical manifold of probability distributions. In standard VAE settings, straight latent paths are not guaranteed to map to meaningful statistical manifold interpolations through the decoder. By regularising the pullback metric of the stochastic decoder, FlatVI induces correspondence between straight paths in the latent space and geodesic interpolations along the manifold of the decoded space.

on a simpler latent manifold with a tractable geometry. (ii) Design a decoding scheme that encourages straight paths in the latent manifold to map to approximate geodesics in the decoded space. (iii) Formalise the geodesic matching framework in a way that supports a flexible choice of the decoder's likelihood. To achieve (i) and (ii), existing methods regularise the latent representation of Gaussian AEs using Euclidean geometry (Chen et al., 2020; Yonghyeon et al., 2021), but limit their application to continuous data by neglecting the decoder's general likelihood model support. Other works explore the connection between stochastic decoders' geometry and the latent space manifold (Arvanitidis et al., 2022), even for discrete data, but do not address regularising the latent manifold to a simple, traversable geometry while preserving geodesic paths on the decoded manifold.

In this work, we close this gap and introduce FlatVI—Flat Variational Inference—a theoretically principled approach pushing straight paths in the latent space of VAEs to approximate geodesic paths along the manifold of the decoded data. Our focus is on statistical manifolds, whose points are probability distributions of a pre-defined family. This enables us to draw connections to the theory of VAEs and information geometry. When trained as a likelihood model, a VAE's decoder image maps latent codes to a statistical manifold's parameter space. This formulation finds direct application in scRNA-seq, where individual gene counts are assumed to follow a negative binomial distribution, reflecting relevant data properties such as discreteness and overdispersion (Zhou et al., 2011).

Crucially, FlatVI regularises the latent space through a *flattening loss* that pushes the pullback metric from a stochastic VAE decoder towards a spatially-uniform, scaled identity matrix, thereby regularising towards a locally Euclidean latent geometry (Figure 1). In a controlled simulation setting, we demonstrate that our regularisation successfully constrains the latent manifold to exhibit an approximate

Euclidean geometry, while enabling likelihood parameter reconstruction on par with standard VAEs. Our method finds direct applications to single-cell representation learning and trajectory inference, which we demonstrate across multiple biological settings by providing an improved data representation for OT-based modelling of cellular population dynamics and latent interpolation. In summary, we make the following contributions:

- We introduce a regularisation technique for discrete-likelihood VAEs to encourage straight latent interpolations to approximate geodesic paths on the statistical manifold induced by the decoder.
- We provide an explicit formulation of the flattening loss for the negative binomial case, which directly impacts modelling high-dimensional scRNA-seq data.
- We empirically validate our model on simulated data and latent geodesic interpolations.
- We show that our method offers a better representation space for existing OT-based trajectory inference tools than existing VAE-based approaches on real data.

## 2. Related Work

**Geometry and AEs.** Prior work by Arvanitidis et al. (2021) introduced optimal latent paths reflecting observation space geometry in deterministic and Gaussian stochastic decoders, extended by Arvanitidis et al. (2022) to VAEs with arbitrary likelihoods. Meanwhile, Chen et al. (2020) explored representation learning benefits by modelling latent spaces of deterministic AEs as flat manifolds, while other studies incorporate data geometry via isometric (Yonghyeon et al., 2021) and Jacobian (Nazari et al., 2023) regularisations.

**Geometry in single-cell representations.** Latent variable models for scRNA-seq data are established (Lopez et al., 2018; Eraslan et al., 2019), and geometric regularisations for continuous approximations of high-dimensional cellular

data have been proposed before for deterministic AEs (Duque et al., 2020; Sun et al., 2025). Combining single-cell representations and geometry, diffusion-based manifold learning (Moon et al., 2019; Huguet et al., 2024; Fasina et al., 2023) offers insights into geometry-aware low-dimensional representations. Investigating single-cell geometry extends to gene expression data (Korem et al., 2015; Qiu et al., 2022) and dynamic settings (Rifkin & Kim, 2002).

**Modelling single-cell state transitions in low-dimensional spaces.** Reconstructing cellular state transitions in a biologically meaningful low-dimensional space is a key challenge in single-cell transcriptomics. Various approaches address this, including applications to drug perturbation prediction (Bunne et al., 2023; Lotfollahi et al., 2023; Hetzel et al., 2022) and multi-modal trajectory inference (Klein et al., 2025). Several works focus on learning continuous gene expression trajectories in latent spaces for modelling cellular dynamics. Our work is tightly linked to Huguet et al. (2022), where the authors employ a Geodesic Autoencoder (GAE) where distances in the latent space approximate geodesic distances in single-cell data, while Haviv et al. (2024) introduce a framework regularising autoencoders' latent spaces to approximate Wasserstein distances, with applications in spatial transcriptomics. Flow Matching (Lipman et al., 2023; Albergo & Vanden-Eijnden, 2023), a generative modelling approach, has also been explored for reconstructing manifold-aware cellular trajectories in low-dimensional spaces (Kapusniak et al., 2024), with OT-based formulations showing promise in this context (Tong et al., 2024a).

## 3. Background

### 3.1. Discrete VAEs for Single-Cell RNA-seq

In this work, we deal with discrete count data, formally collected in a high-dimensional matrix $\mathbf{X} \in \mathbb{N}_0^{N \times G}$, where $N$ represents the number of observations and $G$ the number of features. We assume that individual sample features $x_{ng}$ are independent realisations of a discrete random variable $X_{ng} \sim \mathbb{P}(\cdot | \varphi_{ng})$ with observation-specific real parameters $\varphi_{ng}$. Let $\mathbf{x} \in \mathcal{X} = \mathbb{N}_0^G$ be a single realisation vector.

We consider a joint latent variable model describing the probability of an observation $\mathbf{x}$ and its associated latent variable $\mathbf{z}$. The model factorizes as $p_\phi(\mathbf{x}, \mathbf{z}) = p_\phi(\mathbf{x}|\mathbf{z})p(\mathbf{z})$, where $\mathbf{z} \in \mathcal{Z} = \mathbb{R}^d$ is a $d$-dimensional latent variable with $d < G$, $\mathbf{z} \sim p(\mathbf{z})$, and $p(\mathbf{z}) = \mathcal{N}(\mathbf{0}, \mathbb{I}_d)$. Here, $\mathbb{I}_d$ is the squared identity matrix with dimension $d$.

The factor $p_\phi$ defines a likelihood model following a discrete distribution from a pre-defined family, with parameters expressed as a function of the latent variable $\mathbf{z}$ as:

$$p_\phi(\mathbf{x}|\mathbf{z}) = \mathbb{P}(\mathbf{x}|h_\phi(\mathbf{z})) . \tag{1}$$

In deep latent variable models, $h_\phi$ is a deep neural network

termed *decoder*. VAEs additionally include an *encoder* network $f_\psi : \mathcal{X} \to \mathcal{Z}$ optimized jointly with $h_\phi$ through the Evidence Lower Bound Objective (ELBO) (Kingma & Welling, 2014). Overall, as long as one can select a parametric family of distributions as a reasonable noise model for the dataset properties, the likelihood of the data can be modelled by the decoder of a VAE.

In the field of scRNA-seq, biological and technical variability causes sparsity and overdispersion properties in the expression counts, making the negative binomial likelihood a natural choice for modelling gene expression. *Sparsity* arises from technical limitations in detecting gene transcripts or from unexpressed genes in specific conditions. *Overdispersion* refers to genes having higher variance than the mean, deviating from a Poisson model. This is influenced by technical factors and modelled by the inverse dispersion parameter of a negative binomial distribution (Heumos et al., 2023). Thus, we assume that genes follow a negative binomial noise model $\text{NB}(\mu_{ng}, \theta_g)$, where $\mu_{ng}$ and $\theta_g$ represent the cell-gene-specific mean and the gene-specific inverse dispersion parameters, respectively. In the VAE setting, given a gene-expression vector $\mathbf{x}$, we define the following parameterizations (Lopez et al., 2018):

$$\mathbf{z} = f_\psi(\mathbf{x}), \quad \boldsymbol{\mu} = h_\phi(\mathbf{z}, l) = l \operatorname{softmax}(\rho_\phi(\mathbf{z})) , \tag{2}$$

where $\rho_\phi : \mathbb{R}^d \to \mathbb{R}^G$ models expression proportions of individual genes and $l$ is the observed cell-specific size factor directly derived from the data as a cell's total number of counts $l = \sum_{g=1}^G x_g$. The encoder $f_\psi$ already takes into account the reparametrisation trick (Kingma & Welling, 2014). Assuming global, gene-specific technical effects, $\theta_g$ is treated as a free parameter independent of the cell's state.

### 3.2. The Geometry of Autoencoders

**Continuous deterministic AEs.** A possible assumption is that continuous data lies near a low-dimensional Riemannian manifold $\mathcal{M}_\mathcal{X}$, with $\mathcal{X} = \mathbb{R}^G$, associated with a $d$-dimensional latent space $\mathcal{Z}$. The decoder $h$ of a deterministic AE model can be seen as an immersion $h : \mathbb{R}^d \to \mathbb{R}^G$ of the latent space $\mathcal{Z}$ into the embedded Riemannian manifold $\mathcal{M}_\mathcal{X}$ equipped with a metric tensor $\text{M}$ and defined as follows:

**Definition 3.1.** *A Riemannian manifold is a smooth manifold $\mathcal{M}_\mathcal{X}$ endowed with a Riemannian metric $\text{M}(\mathbf{x})$ for $\mathbf{x} \in \mathcal{M}_\mathcal{X}$. $\text{M}(\mathbf{x})$ is a positive-definite matrix that changes smoothly and defines a local inner product on the tangent space $\mathcal{T}_\mathbf{x}\mathcal{M}_\mathcal{X}$ as $\langle \mathbf{u}, \mathbf{v} \rangle_{\mathcal{M}_\mathcal{X}} = \mathbf{u}^\text{T}\text{M}(\mathbf{x})\mathbf{v}$, with $\mathbf{v}, \mathbf{u} \in \mathcal{T}_\mathbf{x}\mathcal{M}_\mathcal{X}$ (Do Carmo & Flaherty Francis, 1992).*

From Definition 3.1, it follows that a Riemannian manifold in the decoded space has Euclidean geometry if $\text{M}(\mathbf{x}) = \mathbb{I}_G$ everywhere, as the dot product between tangent vectors reduces to a linear product.

In this setting, one can define a Riemannian manifold in the latent space, called $\mathcal{M}_{\mathcal{Z}}$, whose geometry is directly linked to the geometry of the decoded manifold by the *pullback metric* $\mathrm{M}(\mathbf{z})$:

$$\mathrm{M}(\mathbf{z}) = \mathbb{J}_h(\mathbf{z})^{\mathrm{T}} \mathrm{M}(h(\mathbf{z})) \mathbb{J}_h(\mathbf{z}) , \qquad (3)$$

where $\mathbb{J}_h(\mathbf{z})$ is the Jacobian matrix of $h(\mathbf{z})$. Here, the decoder $h$ is assumed to be a diffeomorphism between the latent space and its image, such that $\mathbb{J}_h(\mathbf{z})$ is full rank for all $\mathbf{z}$. The relationship between latent and decoded metric tensors in Equation (3) enables us to connect distances on the latent manifold with quantities measured in the data space. For example, Equation (3) allows to define the shortest curve $\gamma(t)$ connecting pairs of latent codes $\mathbf{z}_1$ and $\mathbf{z}_2$ as the one minimising the distance between their images $h(\mathbf{z}_1)$ and $h(\mathbf{z}_2)$ on $\mathcal{M}_{\mathcal{X}}$. More formally:

$$d_{\text{latent}}(\mathbf{z}_1, \mathbf{z}_2) = \inf_{\gamma(t)} \int_0^1 \|\dot{h}(\gamma(t))\| \mathrm{d}t \qquad (4)$$

$$= \inf_{\gamma(t)} \int_0^1 \sqrt{\dot{\gamma}(t)^{\mathrm{T}} \mathrm{M}(\gamma(t)) \dot{\gamma}(t)} \mathrm{d}t , \quad (5)$$

$$\text{where} \quad \gamma(0) = \mathbf{z}_1, \ \gamma(1) = \mathbf{z}_2 .$$

Here, $\gamma(t) : \mathbb{R} \to \mathcal{Z}$ is a curve in the latent space with boundary conditions $\gamma(0) = \mathbf{z}_1$ and $\gamma(1) = \mathbf{z}_2$, and $\dot{\gamma}(t)$ its derivative along the manifold (more details in Appendix C.1). Importantly, when the metric tensor satisfies $\mathrm{M}(\mathbf{z}) = \mathbb{I}_d$ for all $\mathbf{z}$, the curve $\gamma^\star(t)$ minimising Equation (4) is the straight line between latent codes, and the geodesics coincide with Euclidean lines in latent space.

**Stochastic decoders.** While in AEs one deals with deterministic manifolds, in VAEs the decoder function $h$ maps a latent code $\mathbf{z} \in \mathcal{Z}$ to the parameter configuration $\boldsymbol{\varphi} \in \mathcal{H}$ of the data likelihood. If the likelihood has continuous parameters, $\mathcal{H} = \mathbb{R}^G$ represents the parameter space. As such, the *image of the decoder lies on a statistical manifold*, which is a smooth manifold of probability distributions. Such manifolds have a natural metric tensor called *Fisher Information Metric* (FIM) (Nielsen, 2020; Arvanitidis et al., 2022). The FIM defines the local geometry of the statistical manifold and can be used to build the pullback metric for arbitrary decoders. For a statistical manifold $\mathcal{M}_{\mathcal{H}}$ with parameters $\boldsymbol{\varphi} \in \mathcal{H}$, the FIM is formulated as

$$\mathrm{M}(\boldsymbol{\varphi}) = \mathbb{E}_{p(\mathbf{x}|\boldsymbol{\varphi})} \left[ \nabla_{\boldsymbol{\varphi}} \log p(\mathbf{x}|\boldsymbol{\varphi}) \nabla_{\boldsymbol{\varphi}} \log p(\mathbf{x}|\boldsymbol{\varphi})^{\mathrm{T}} \right] , \quad (6)$$

where $\boldsymbol{\varphi} = h(\mathbf{z})$ and the metric tensor $\mathrm{M}(\boldsymbol{\varphi}) \in \mathbb{R}^{G \times G}$. Analogous to deterministic AEs, one can combine Equation (6) and Equation (3) to formulate the pullback metric for an arbitrary statistical manifold, with the difference that the metric tensor is defined based on the parameter space $\mathcal{H}$. Thus, the latent space of a VAE is endowed with the pullback metric for a statistical manifold.

$$\mathrm{M}(\mathbf{z}) = \mathbb{J}_h(\mathbf{z})^{\mathrm{T}} \mathrm{M}(\boldsymbol{\varphi}) \mathbb{J}_h(\mathbf{z}) , \qquad (7)$$

where $\mathrm{M}(\mathbf{z}) \in \mathbb{R}^{d \times d}$. Note that the calculation of the FIM is specific for the chosen likelihood type and depends on initial assumptions on the data distribution (see in Appendix C.2).

## 4. The FlatVI Model

### 4.1. Latent Euclidean Assumption in Single-Cell Biology

Modelling high-dimensional cellular processes from discrete count data poses significant challenges. A common approach is to study variations in cell states as latent interpolations using the continuous latent representation learned by negative binomial VAEs. In applications like perturbation modelling (Hetzel et al., 2022; Lotfollahi et al., 2023) or gene expression distance quantification (Luecken et al., 2021), it is a common assumption to model state transitions as linear shifts in the latent space. This also applies to trajectory inference for modelling *continuous* population dynamics, where the trajectory of single cells is learnt by matching subsequent cellular snapshots collected over time using dynamic OT in Euclidean spaces (Tong et al., 2020; Koshizuka & Sato, 2023; Tong et al., 2024b; Neklyudov et al., 2023). In all the above cases, the standard assumption is that linear interpolations of the latent manifold reflect optimal trajectories on the decoded single-cell manifold. However, the standard negative binomial VAE formulation *does not naturally enforce such a correspondence*, violating modelling assumptions and potentially leading to sub-optimal decoded cell-state trajectories and latent distance estimations.

To address this issue and complement existing methods with representations meeting their assumptions, we propose to regularise the latent space of a single-cell VAE in such a way that the latent manifold $\mathcal{M}_{\mathcal{Z}}$ has an approximately Euclidean geometry. In other words, *our goal is to encourage correspondence between straight paths in the latent space and geodesic interpolations along the statistical manifold.* When this condition is satisfied, decoded trajectories generated using linear interpolations in the latent space of the VAE respect the geometry of the data manifold.

### 4.2. Assumptions

Before describing our regularisation approach, we state two assumptions about the geometry of the single-cell manifold induced by the negative binomial decoder:

1. **Geodesic convexity:** The manifold is assumed to be geodesically convex; that is, any two points on the manifold are connected by a unique geodesic.

2. **Local-to-global approximation:** The manifold is sufficiently sampled such that enforcing local geometric constraints via the pullback metric at observed points yields a good approximation of the global geometry.

Assumption (1) is appropriate for acyclic biological pro-

cesses, such as differentiation or perturbation responses, where cellular transitions follow smooth and directed progressions. Assumption (2) aligns with standard practice in single-cell manifold learning, where local neighbourhood structure is leveraged to infer global geometry under the assumption of smooth state transitions.

### 4.3. Flattening Loss

To ensure that straight latent paths approximate geodesics on the decoded statistical manifold, we introduce a regularisation term in the VAE objective that introduces a locally Euclidean geometry in the latent manifold $\mathcal{M}_{\mathcal{Z}}$.

As shown in Section 3.2, the local geometry of a VAE's latent space is determined by the metric in Equation (7). This metric is a function of the Fisher information of the decoder's likelihood, which depends on the decoded parameters $\varphi \in \mathcal{H}$. From Equation (4) we also know that if $\mathrm{M}(\mathbf{z}) = \mathbb{I}_d$, then the geodesic distance between each pair of latent points is given by the straight line between them. Therefore, regularising the product $\mathbb{J}_h(\mathbf{z})^{\mathrm{T}}\mathrm{M}(\varphi)\mathbb{J}_h(\mathbf{z})$ towards $\mathbb{I}_d$ forces a VAE to model locally Euclidean latent geometry. Crucially, the non-linear decoder is still trained to reconstruct the original data space under the likelihood optimisation task in the ELBO, preserving the local geometry of the decoded statistical manifold described by $\mathrm{M}(\varphi)$.

In summary, we introduce a *flattening loss*, $\mathcal{L}_{\text{flat}}$, which encourages locally Euclidean latent geometry in VAEs with flexible decoders. This loss is combined with the ELBO to form the full FlatVI objective:

$$\mathcal{L}_{\text{flat}}(\phi, \psi, \alpha) = \mathbb{E}_{q_\psi(\mathbf{z}|\mathbf{x})} \|\mathrm{M}(\mathbf{z}) - \alpha\mathbb{I}_d\|_F^2 \ . \qquad (8)$$

Here, $\phi$ and $\psi$ are the VAE's parameters, $q$ the approximate posterior on the latent space learnt by the encoder, and $\mathrm{M}(\mathbf{z})$ is calculated by Equation (7). Meanwhile, $\alpha$ is a trainable parameter offering some flexibility on the scale of the diagonal constraint while preserving straight geodesics. The Frobenius norm encourages each local pullback metric to be close to a scaled identity. In VAEs, the loss of FlatVI is combined with the ELBO:

$$\mathcal{L}_{\text{FlatVI}}(\phi, \psi, \alpha) = \mathcal{L}_{\text{ELBO}}(\phi, \psi) + \lambda\mathcal{L}_{\text{flat}}(\phi, \psi, \alpha) \ , \quad (9)$$

where $\lambda$ controls the strength of the flattening regularisation. We summarise the procedure used to train FlatVI in Algorithm 1.

### 4.4. FlatVI on a Negative Binomial Single-Cell Manifold

In this work, we model cellular trajectories to study the evolution of biological processes through interpolations on a statistically grounded manifold. As outlined in Section 3.1, single-cell counts are modelled with a negative binomial decoder with the following univariate point mass function

for each gene $g$ independently:

$$p_{\text{NB}}(x_g|\mu_g, \theta_g) = C \left(\frac{\theta_g}{\theta_g + \mu_g}\right)^{\theta_g} \left(\frac{\mu_g}{\theta_g + \mu_g}\right)^{x_g}, \quad (10)$$

$$\text{where } C = \frac{\Gamma(\theta_g + x_g)}{x_g!\Gamma(\theta_g)} \ , \ \ \mu, \theta > 0$$

with $x_g \in \mathbb{N}_0$ and $\mu_g = h_g(\mathbf{z})$. Notably, since the decoder $h$ produces cell-specific means, each cell is deemed as an individual probability distribution. Consequently, we assume that the single-cell data lies on a statistical manifold parameterised by the decoder in the space of negative binomial distributions. According to Equation (7), we pull back the FIM of the statistical manifold of the negative binomial probability distribution to the latent manifold $\mathcal{M}_{\mathcal{Z}}$.

**Proposition 4.1.** *The pullback metric at a latent point* $\mathbf{z} \in \mathcal{Z}$ *of the statistical manifold of negative binomial distributions, parameterised by a decoder* $h$ *and fixed inverse dispersion* $\boldsymbol{\theta}$*, is given by:*

$$\mathrm{M}(\mathbf{z}) = \sum_g \frac{\theta_g}{h_g(\mathbf{z})(h_g(\mathbf{z}) + \theta_g)} \nabla_{\mathbf{z}} h_g(\mathbf{z}) \otimes \nabla_{\mathbf{z}} h_g(\mathbf{z}) \ ,$$
$$(11)$$

*where* $\otimes$ *is the outer product of vectors, g indexes individual decoded dimensions, and* $h_g(\mathbf{z})$ *denotes the decoded mean for gene g.*

We provide the derivation of Proposition 4.1 in Appendix B. Note that we only take the gradient of the mean decoder $h$ since the inverse dispersion parameter is not a function of the latent space in single-cell VAEs (see Section 3.1). This expression for $\mathrm{M}(\mathbf{z})$ is then used in the flattening loss (Equation (8)) to train a geometry-regularised single-cell VAE.

## 5. Experiments

We evaluate FlatVI on both simulated and real single-cell scenarios. We begin in Section 5.1 by demonstrating that our regularisation improves the approximation of constant Euclidean geometry in the latent manifold while preserving the reconstruction of likelihood parameters on synthetic data. For real-world validation, we focus on modelling single-cell population dynamics using linear dynamic OT. We investigate whether enforcing Euclidean geometry in the latent space improves both latent and gene-wise trajectory reconstruction. These results are presented in Section 5.2 and Section 5.3. Finally, in Section 5.4 and Section 5.5, we explore the representations learned by FlatVI and evaluate the effectiveness of linear latent interpolations for modelling developmental processes.

### 5.1. Simulated Data

**Task and datasets.** To assess the effect of FlatVI on latent geometry, we evaluate its performance on a synthetic dataset

*Table 1.* Comparison between FlatVI and an unregularised NB-VAE ($\lambda = 0$) in terms of MSE for $\boldsymbol{\mu}$ and $\boldsymbol{\theta}$, and the 3-NN overlap between Euclidean and geodesic neighbourhoods.

| Reg. strength $\lambda$ | MSE ($\boldsymbol{\mu}$) ($\downarrow$) | MSE ($\boldsymbol{\theta}$) ($\downarrow$) | 3-NN overlap ($\uparrow$) |
|:---:|:---:|:---:|:---:|
| $\lambda = 0$ | $15.52_{\pm 0.94}$ | $3.10_{\pm 0.19}$ | $0.66_{\pm 0.00}$ |
| $\lambda = 1$ | $16.34_{\pm 0.46}$ | $5.67_{\pm 0.88}$ | $0.63_{\pm 0.00}$ |
| $\lambda = 3$ | $16.35_{\pm 0.53}$ | $3.09_{\pm 0.31}$ | $0.77_{\pm 0.00}$ |
| $\lambda = 5$ | $\mathbf{14.75}_{\pm 0.12}$ | $3.20_{\pm 0.20}$ | $0.67_{\pm 0.01}$ |
| $\lambda = 7$ | $15.47_{\pm 0.20}$ | $3.38_{\pm 0.09}$ | $0.72_{\pm 0.01}$ |
| $\lambda = 10$ | $15.41_{\pm 0.07}$ | $\mathbf{3.08}_{\pm 0.13}$ | $\mathbf{0.80}_{\pm 0.03}$ |

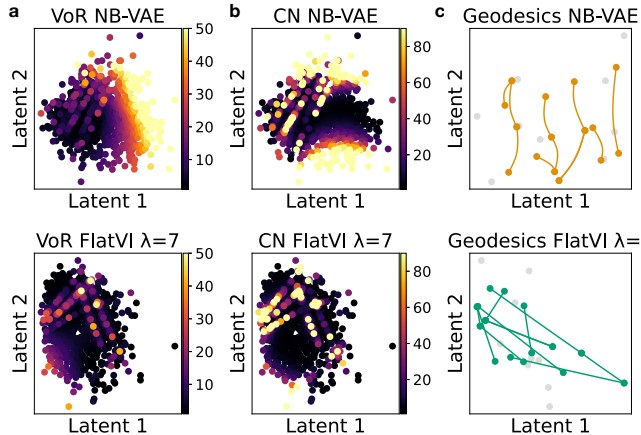

*Figure 2.* Comparison of latent geometries for NB-VAE (top) and FlatVI with $\lambda = 7$ (bottom) using: **(a)** Variance of the Riemannian metric (VoR), **(b)** Condition Number (CN), and **(c)** Straightness of geodesics between high VoR regions.

generated from a multivariate negative binomial distribution. We aim to induce Euclidean latent geometry in a 2D space while maintaining faithful data reconstruction.

We simulate 1000 observations from a 10-dimensional negative binomial distribution with known mean ($\boldsymbol{\mu}$) and inverse dispersion ($\boldsymbol{\theta}$). Each observation represents a simulated cell, uniformly assigned to one of three categories mimicking biological cell types (see Figure 6). For each type, the mean vectors are sampled from normal distributions centred at $-1$, $0$, and $1$, respectively, with a standard deviation of $1$, and exponentiated to ensure positivity. Gene-specific inverse dispersion parameters are sampled from a Gamma distribution (concentration 2, rate 1) and made positive. To comply with assumptions in real single-cell data, all data points share the same dispersion values. We provide further details on the simulation setup in Appendix I.2.

**Evaluation.** For $\lambda \in \{0, 1, 3, 5, 7, 10\}$, we assess: (i) The MSE in reconstructing the true $\boldsymbol{\mu}$ and $\boldsymbol{\theta}$ from decoded latent variables; and (ii) the 3-Nearest-Neighbour (3-NN) overlap between neighbourhoods defined by Euclidean and pullback geodesic distances in the latent space (see Appendix L.1.3 for results using more neighbours). Geodesics are approximated by parameterised cubic splines minimising the length under the pullback metric defined in Equation (7) (see Equation (4)). In our setting, a successful regularisation recovers the true parameters of the data-generating process while imposing a similar neighbourhood structure between latent Euclidean and pullback geodesic distances.

We also visually assess Riemannian characteristics of the latent space: The *variance of the Riemannian metric* (VoR) and the *condition number* (CN), following Chen et al. (2020) and Yonghyeon et al. (2021). VoR quantifies how much the metric deviates from its spatial average $\bar{\mathrm{M}} = \mathbb{E}_{\mathbf{z} \sim p_{\mathbf{z}}}[\mathrm{M}(\mathbf{z})]$. A VoR of 0 implies a globally uniform metric, and we estimate its value using batches of 256 latent encodings. CN, defined as the ratio of the largest to the smallest eigenvalue of $\mathrm{M}(\mathbf{z})$, approaches 1 when the metric resembles the identity. Low VoR and CN values indicate a well-flattened latent space. See Appendix I.1 for definitions.

**Results.** In our simulation setting, results in Table 1 show

that our regularisation forces geodesic distances under the pullback metric to better approximate Euclidean topology in the latent space compared to an unregularised Negative Binomial VAE (NB-VAE) with $\lambda = 0$. In other words, increasing $\lambda$ improves the correspondence between the neighbourhood structures induced by pullback geodesic and Euclidean distances (see Appendix L.1.3 for additional metrics). Meanwhile, the capabilities of our model to reconstruct the mean ($\boldsymbol{\mu}$) and inverse dispersion ($\boldsymbol{\sigma}$) parameters do not degrade when the regularisation strength is increased.

The plots in Figure 2 serve as additional proof of the flattening mechanism. Inducing Euclidean geometry into the latent space ensures a more uniform local geometry, as the latent manifold of FlatVI does not exhibit as many regions of systematically high VoR or CN as in the standard NB-VAE setting (see Figure 2a-b). Despite the flattening, some limited regions with high CN and VoR remain in the FlatVI embedding. We qualitatively investigate the cause for high VoR and CN values in Appendix L.1.4.

In Figure 2c we sample 10 couples of points from regions of high VoR in the NB-VAE latent space and plot geodesic paths approximated according to Equation (4) on both FlatVI and the unregularised model. FlatVI achieves straight paths, while pullback-based geodesic interpolations in the standard NB-VAE bottleneck show a curvature (see Figure 7 for comparison with a Euclidean manifold). These findings support our objective of ensuring that pullback-driven geodesics exhibit linear behaviour in the latent space.

### 5.2. Reconstruction of scRNA-seq Trajectories

**Task and dataset.** Our core hypothesis is that FlatVI's latent space provides a better embedding for cell-state

*Table 2.* Comparison of cellular trajectory reconstruction on held-out time points using different representation models. Latent trajectories are learnt with OT-CFM, leaving out intermediate time points and using them as ground truth for evaluating the interpolation of the cellular dynamics. Distribution matching metrics are evaluated to compare real held-out points and reconstructions thereof in both the decoded and latent space across three seeds.

| | EB | | | | MEF | | | |
|---|---|---|---|---|---|---|---|---|
| | Latent | | Decoded | | Latent | | Decoded | |
| | 2-Wasserstein ($\downarrow$) | $L^2$ ($\downarrow$) | 2-Wasserstein ($\downarrow$) | MMD ($\downarrow$) | 2-Wasserstein ($\downarrow$) | $L^2$ ($\downarrow$) | 2-Wasserstein ($\downarrow$) | MMD ($\downarrow$) |
| GAE | $2.16_{\pm 0.14}$ | $0.40_{\pm 0.06}$ | $70.29_{\pm 0.00}$ | $0.14_{\pm 0.04}$ | $2.49_{\pm 0.22}$ | $0.57_{\pm 0.07}$ | $106.83_{\pm 0.01}$ | $0.38_{\pm 0.01}$ |
| NB-VAE | $2.07_{\pm 0.07}$ | $0.30_{\pm 0.02}$ | $43.36_{\pm 0.19}$ | $0.09_{\pm 0.01}$ | $2.07_{\pm 0.12}$ | $0.40_{\pm 0.05}$ | $103.29_{\pm 0.01}$ | $0.19_{\pm 0.01}$ |
| FlatVI | $\mathbf{1.54_{\pm 0.09}}$ | $\mathbf{0.27_{\pm 0.03}}$ | $\mathbf{41.99_{\pm 0.04}}$ | $\mathbf{0.07_{\pm 0.01}}$ | $\mathbf{1.64_{\pm 0.13}}$ | $\mathbf{0.36_{\pm 0.05}}$ | $\mathbf{97.12_{\pm 0.01}}$ | $\mathbf{0.16_{\pm 0.01}}$ |

interpolation methods that assume Euclidean geometry, as our flattening loss encourages the VAE to approximate local Euclidean geometry in the latent manifold. We demonstrate the benefits of using FlatVI's representation space in combination with Euclidean, continuous OT to map single-cell trajectories over time. As a dynamic OT algorithm, we use the OT Conditional Flow Matching (OT-CFM) model (Tong et al., 2024a), which leverages straight-line interpolation between samples to learn a velocity field transporting cells across time (Appendix D).

We evaluate using two real-world datasets: (i) The Embroid Body (EB) dataset (Moon et al., 2019), comprising 18,203 differentiating human embryoid cells over five time points and spanning four lineages; and (ii) the MEF reprogramming dataset (Schiebinger et al., 2019), containing 165,892 cells across 39 time points, tracing the reprogramming of mouse embryonic fibroblasts into induced pluripotent stem cells. Full dataset details are provided in Appendix J.1.

**Baselines.** We compare FlatVI with a standard NB-VAE trained without regularisation (Lopez et al., 2018) as a representation model for continuous OT. Additionally, we evaluate latent OT on embeddings produced by the GAE model from Huguet et al. (2022), described in Section 2. This model is trained on log-normalised gene expression to compensate for the absence of a discrete probabilistic decoder. Differences between FlatVI and GAE are discussed in Appendix E.1.2. All three approaches are used to generate latent embeddings of time-resolved gene expression datasets. These embeddings are then used to train an OT-CFM model that learns latent trajectories from unpaired observations at consecutive time points.

**Evaluation.** Following Tong et al. (2020), we leave out intermediate time points during training and assess the model's ability to reconstruct them via OT. The accuracy of reconstructing an unseen time point $t$ from $t-1$ reflects the model's interpolation ability along the data manifold. We use this paradigm to compare different representation spaces. For quantitative evaluation, we compute the 2-Wasserstein and mean $L^2$ distances between real and reconstructed latent cells at each time point. We also assess decoded gene expres-

sion quality using linear-kernel Maximum Mean Discrepancy (MMD) (Borgwardt et al., 2006) and 2-Wasserstein distance.

We set FlatVI's regularisation strength to $\lambda=1$ for the EB dataset and $\lambda=0.1$ for the MEF dataset. We tune the hyperparameter based on the value that leads to the best representation for OT-based trajectory reconstruction on training data (see Appendix H for more details).

**Results.** Table 2 reports reconstruction metrics between true and interpolated latent cells. Across all datasets and metrics, trajectories in FlatVI's Euclidean latent space result in better time point reconstruction compared to the baseline models, highlighting the effectiveness of our regularisation. Furthermore, FlatVI improves the quality of decoded gene expression trajectories, as reflected in lower MMD and Wasserstein distances. Table 6 additionally provides biological validation via improved reconstruction of lineage marker trajectories using FlatVI.

### 5.3. Latent Vector Field and Lineage Mapping

**Task and dataset.** We evaluate the capacity of continuous OT to identify a biologically meaningful cell velocity field using the representation spaces computed by FlatVI, the unregularised NB-VAE and the GAE model. We hypothesise that applying dynamic OT with Euclidean cost to a flat representation space is beneficial. As a dataset for the analysis, we employ the pancreatic endocrinogenesis (here shortly denoted as Pancreas) by Bastidas-Ponce et al. (2019), which measures 16,206 cells and spans embryonic days 14.5 to 15.5, revealing multipotent cell differentiation into endocrine and non-endocrine lineages. More specifically, we train the compared representation learning frameworks on the dataset and learn separate vector fields for all models' embeddings, matching days 14.5 to 15.5 with OT-CFM. The learnt vector field represents the directionality of the observations on the cellular development manifold.

**Evaluation.** Using the CellRank model (Lange et al., 2022; Weiler et al., 2024), we build random walks on a cell graph based on the directionality of latent velocities learnt by OT-CFM in the different representation spaces.

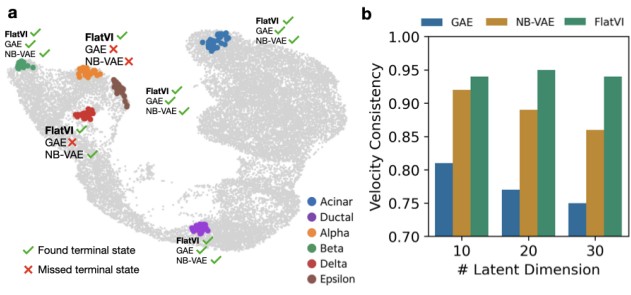

*Figure 3.* Learning terminal states from OT-CFM's cell velocities in the Pancreas dataset. (**a**) Terminal states found by CellRank using different representation models. (**b**) Latent velocity consistency computed for cells across different latent space sizes.

Walks converge to macrostates representing the endpoints of the biological process if the learnt velocity field points to biologically meaningful directions. We quantify the quality of vector fields learnt by OT in different latent spaces based on (i) the number of macrostates identified by random walks, and (ii) the velocity consistency, measured as the correlation of the latent velocity field of single datapoints with that of the neighbouring cells (Gayoso et al., 2024). Higher consistency indicates smoother transitions in the vector field, suggesting that the representation space facilitates more coherent and biologically meaningful dynamics, making it a suitable space for learning trajectories (see Appendix I.1).

**Results.** Figure 3a summarises the number of terminal cell states identified by following the velocity graph. From prior biological knowledge, it is known that the dataset contains six terminal states, which are all identified on the representation computed by our FlatVI ($\lambda = 1$). In contrast, on the GAE and NB-VAE's representations, CellRank only captures four and five terminal states, respectively. In Figure 3b, we further evaluate the velocity consistency within neighbourhoods of cells as a function of latent dimensionality. In line with previous results, OT on the approximately Euclidean latent space computed by FlatVI yields a more consistent velocity field across latent space sizes.

## 5.4. Single-Cell Data Representations

We visualise single-cell latent representations on the previously introduced datasets computed using the FlatVI, NB-VAE, and GAE models. For FlatVI, the value of $\lambda$ is set to 1 for EB and Pancreas and 0.1 for the MEF dataset, in line with previous settings. In Figure 4, we compare the Principal Component (PC) embeddings of FlatVI's latent space with competing models, highlighting initial and terminal cellular states. Despite the regularisation, FlatVI represents the biological structure in the latent space better or on par with the baselines, as illustrated by the separation between initial and terminal states. This is particularly evident in the MEF dataset, where FlatVI provides a clearer division

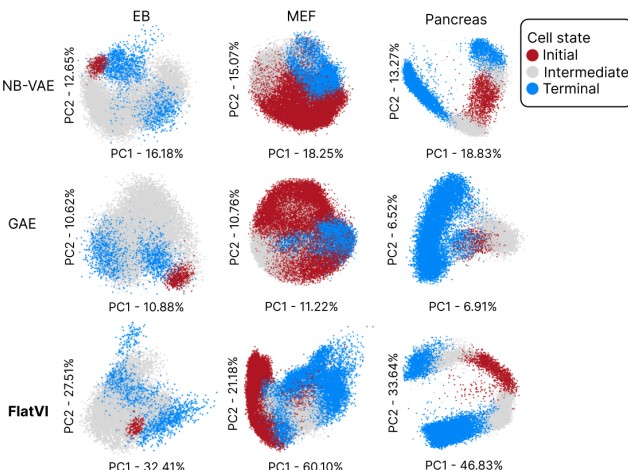

*Figure 4.* 2D PCA plots of the latent spaces computed by GAE, NB-VAE and FlatVI. Marked are initial, intermediate and terminal cell states along the biological trajectory.

between initial and terminal cell types. In Table 8 we show that such a separation is more pronounced than competing models, also on the Pancreatic dataset based on quantitative clustering metrics. Moreover, the higher variance explained by individual PCs in FlatVI's latent space suggests that our model captures the main sources of variation (the biological trajectories) more efficiently, reducing latent space dimensionality and enhancing information compression while preserving or even improving biological fidelity. Consequently, FlatVI is well-suited for smaller latent spaces, making it a promising input for trajectory inference methods.

## 5.5. Decoded Geodesic Interpolations

**Task and evaluation.** We qualitatively evaluate whether linear interpolations in the latent space of FlatVI yield biologically meaningful trajectories when decoded into gene expression space. Given two cells at different stages along a developmental lineage, we linearly interpolate between their latent representations and decode the intermediate points. We then inspect whether the expression of known marker genes along these decoded paths evolves consistently with expected biological progression. If so, this suggests that linear paths in FlatVI's latent space approximate geodesics aligned with the underlying developmental manifold.

**Baseline.** As a baseline, we use GAGA (Sun et al., 2025), which explicitly enforces alignment between the latent geometry and data structure using neighbour-based regularisation. Unlike FlatVI, GAGA does not assume a tractable latent manifold. Instead, it learns a data-driven representation where geodesics are approximated via a neural ODE. In contrast, FlatVI enables efficient approximations of latent geodesics through simple linear interpolation.

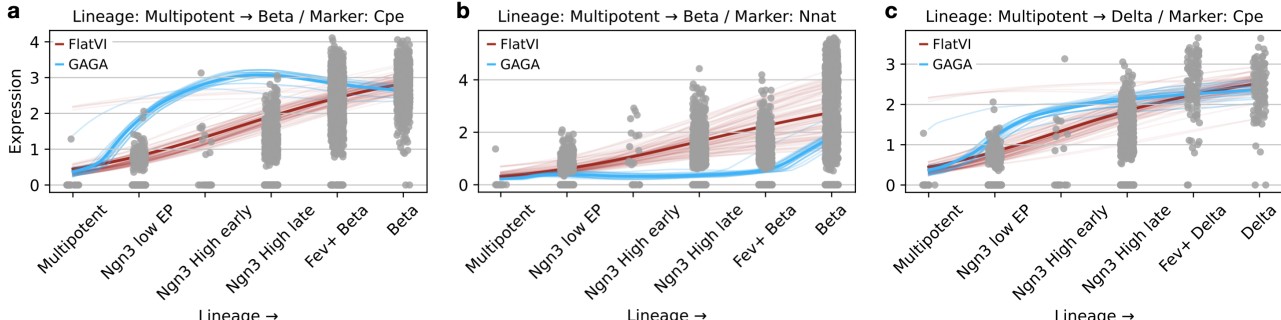

*Figure 5.* Decoded marker trajectories from latent interpolations on the Pancreas dataset, comparing FlatVI and GAGA. We sample 100 pairs of multipotent and mature cells from each terminal state, encode their gene expression profiles, and perform latent interpolations (linearly in FlatVI and using a neural ODE in GAGA). Intermediate states along these trajectories are decoded, and the resulting marker gene expression is visualised alongside the real expression along the lineage (dark grey points). Each trajectory is shown as a low-opacity line, while the solid line denotes the mean trajectory. We show three examples (**a**, **b**, and **c**) from the Beta and Delta lineages.

**Dataset.** We use the pancreas endocrinogenesis dataset described in Section 5.3, focusing on the endocrine developmental trajectory. We randomly sample batches of 100 multipotent progenitor cells and 100 terminal-stage cells from each of the endocrine branch lineages. Pairs of multipotent and mature cells are randomly matched, and their decoded interpolation paths are analysed for marker gene expression dynamics.

**Results.** While both approaches produce reasonable results on average (see Figure 13), in Figure 5 we highlight some recurrent sub-optimal patterns in GAGA's performance that do not arise in FlatVI. Specifically, in Figure 5 we present examples of unstable optimisation (Figure 5a), underestimation of the marker expression (Figure 5b) and overestimation of intermediate expression patterns (Figure 5c). In contrast, FlatVI's latent space consistently produces decoded marker dynamics that faithfully recapitulate the expected fate-specific trajectories. Reasonably, interpolations computed with FlatVI are less computationally expensive (Figure 14).

These findings highlight the effectiveness of FlatVI as a straightforward and reliable method for exploring cellular manifolds, without requiring a parameterised interpolant.

## 6. Conclusion

We addressed the problem of modelling cellular trajectories in scRNA-seq data by introducing FlatVI, a VAE training strategy that enforces a locally Euclidean geometry in the latent space by regularising the pullback metric of the stochastic decoder. This regularisation encourages straight latent paths to approximate geodesic interpolations in the decoded data space. Experiments on synthetic data demonstrate that FlatVI successfully induces a latent Euclidean geometry while preserving accurate parameter reconstruction. When combined with dynamic OT, FlatVI improves trajectory prediction performance and yields more consistent vector fields

on cellular manifolds. Furthermore, linear interpolations between latent cellular states offer interpretable insights into the progression of cell states, providing a straightforward approach to exploring dynamical biological processes. Collectively, these improvements enhance core tasks in cellular development, such as fate mapping and the reconstruction of differentiation pathways, establishing FlatVI as a useful tool for trajectory inference in single-cell transcriptomics.

**Limitations and future work.** To improve FlatVI's applicability to real-world datasets, we aim to enhance its robustness to reconstruction loss on biological data, reducing potential trade-offs between flattening and reconstruction likelihood. As discussed in Section 4, enforcing a locally Euclidean latent geometry imposes strong assumptions that may not hold for all datasets (e.g., those dominated by cyclic processes such as the cell cycle). Future work will investigate alternative latent geometries to better capture diverse biological structures. We also plan to extend FlatVI to a broader class of statistical manifolds and single-cell tasks, including Poisson-based modelling of chromatin accessibility, evaluation in batch correction settings, and OT-mediated perturbation modelling.

## Impact Statement

The presented work deals with fundamental characteristics of scRNA-seq data and studies how efficient representations of complex high-dimensional cellular data can help to address key biological questions. We envision the release of FlatVI as a user-friendly, open-source tool to enable its widespread use as an option for single-cell analysis. Dealing with biological data, FlatVI could be used in sensitive settings involving clinical information and patient data.

## Acknowledgments

The authors give special thanks to Sören Becker for providing valuable feedback on the manuscript. Additionally, A.P. and L.H. are supported by the Helmholtz Association under the joint research school Munich School for Data Science (MUDS). A.P., S.G. and F.J.T. also acknowledge support from the German Federal Ministry of Education and Research (BMBF) through grant numbers 031L0289A, 031L0289C, 01IS18053A and 031L0269C. F.J.T. acknowledges support from the Helmholtz Association's Initiative and Networking Fund via the CausalCellDynamics project (grant number Interlabs-0029). Finally, F.J.T. acknowledges support from the European Union (ERC, DeepCell - grant number 101054957 and ERA_PerMed-NET Transcan-3 - grant agreement number 01KT2308).

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

## A. Code and Datasets

We have made our code publicly available at https://github.com/theislab/FlatVI. All datasets used in this study are open source, and their associated publications are cited in the manuscript.

## B. Derivation of the Fisher Information Metric for the Negative Binomial Distribution

We first show that the Fisher information of a univariate Negative Binomial (NB) distribution parameterised by the mean $\mu$ and inverse dispersion $\theta$ with respect to $\mu$ is

$$\mathrm{M}(\mu) = \frac{\theta}{\mu(\mu + \theta)} . \tag{12}$$

We then move on with the derivation of the pullback metric in Proposition 4.1.

**Fisher information of the NB distribution.** The univariate NB probability distribution parameterised by mean $\mu$ and inverse dispersion $\theta$ is

$$p_{\mathrm{NB}}(x \mid \mu, \theta) = \frac{\Gamma(\theta + x)}{x!\Gamma(\theta)} \left(\frac{\theta}{\theta + \mu}\right)^{\theta} \left(\frac{\mu}{\theta + \mu}\right)^{x} . \tag{13}$$

The Fisher information of the distribution can be computed with respect to $\mu$ as:

$$\mathrm{M}(\mu) = -\mathbb{E}_{p(x|\mu,\theta)} \left[ \frac{\partial^2}{\partial \mu^2} \log p_{\mathrm{NB}}(x \mid \mu, \theta) \right] , \tag{14}$$

where

$$\log p_{\mathrm{NB}}(x \mid \mu, \theta) = C + \theta \left[\log(\theta) - \log(\theta + \mu)\right] + x \left[\log(\mu) - \log(\theta + \mu)\right] , \tag{15}$$

with $C = \log(\Gamma(\theta + x)) - \log(x!) - \log(\Gamma(\theta))$. Then, it can be shown that

$$\frac{\partial^2}{\partial \mu^2} \log p_{\mathrm{NB}}(x \mid \mu, \theta) = \frac{\theta + x}{(\theta + \mu)^2} - \frac{x}{\mu^2} . \tag{16}$$

Using the fact that the parameterisation involving the mean $\mu$ and inverse dispersion $\theta$ implies that

$$\mathbb{E}_{p(x|\mu,\theta)} [x] = \mu , \tag{17}$$

we can expand Equation (14) as follows

$$\begin{aligned}
\mathrm{M}(\mu) &= -\mathbb{E}_{p(x|\mu,\theta)} \left[ \frac{\theta + x}{(\theta + \mu)^2} - \frac{x}{\mu^2} \right] \\
&= -\frac{1}{(\theta + \mu)^2} \mathbb{E}_{p(x|\mu,\theta)} [\theta + x] + \frac{1}{\mu^2} \mathbb{E}_{p(x|\mu,\theta)} [x] \\
&= \frac{\theta}{\mu(\mu + \theta)} .
\end{aligned} \tag{18}$$

**Derivation of the Fisher information metric.** We here consider the NB-VAE case, where the likelihood is parameterised by $\mu_g = h_g(\mathbf{z})$ and $\theta_g$ independently for each gene $g$.

When $h$ is a continuously differentiable function of $\mathbf{z}$, the pullback metric $\mathrm{M}_g(\mathbf{z})$ of the output $g$ w.r.t $\mathbf{z}$ by the reparameterisation property (Lehmann & Casella, 2006) is

$$\begin{aligned}
\mathrm{M}_g(\mathbf{z}) &= \nabla_{\mathbf{z}} h_g(\mathbf{z}) \mathrm{M}(h_g(\mathbf{z})) \nabla_{\mathbf{z}} h_g(\mathbf{z})^T \\
&= \frac{\theta_g}{h_g(\mathbf{z})(h_g(\mathbf{z}) + \theta_g)} \nabla_{\mathbf{z}} h_g(\mathbf{z}) \otimes \nabla_{\mathbf{z}} h_g(\mathbf{z}) ,
\end{aligned} \tag{19}$$

where $\otimes$ is the outer product of vectors, and the gradients are column vectors.

By the chain rule, the joint Fisher information of independent random variables equals the sum of the Fisher information values of each variable (Zamir, 1998). As all $x_g$ are independent given $\mathbf{z}$ in the NB-VAE, the resulting Fisher Information Metric (FIM) is

$$\mathrm{M}(\mathbf{z}) = \sum_g \mathrm{M}_g(\mathbf{z})$$

$$= \sum_g \frac{\theta_g}{h_g(\mathbf{z})(h_g(\mathbf{z}) + \theta_g)} \nabla_{\mathbf{z}} h_g(\mathbf{z}) \otimes \nabla_{\mathbf{z}} h_g(\mathbf{z}) . \tag{20}$$

## C. The Geometry of AEs

We deal with the assumption that the observed data lies near a Riemannian manifold $\mathcal{M}_{\mathcal{X}}$ embedded in the ambient space $\mathcal{X} = \mathbb{R}^G$. The manifold $\mathcal{M}_{\mathcal{X}}$ is defined as follows:

**Definition C.1.** A Riemannian manifold is a smooth manifold $\mathcal{M}_{\mathcal{X}}$ endowed with a Riemannian metric $\mathrm{M}(\mathbf{x})$ for $\mathbf{x} \in \mathcal{M}_{\mathcal{X}}$. $\mathrm{M}(\mathbf{x})$ changes smoothly and identifies an inner product on the tangent space $\mathcal{T}_{\mathbf{x}}\mathcal{M}_{\mathcal{X}}$ at a point $\mathbf{x} \in \mathcal{M}_{\mathcal{X}}$ as $\langle \mathbf{u}, \mathbf{v} \rangle_{\mathcal{M}_{\mathcal{X}}} = \mathbf{u}^{\mathbf{T}}\mathrm{M}(\mathbf{x})\mathbf{v}$, with $\mathbf{v}, \mathbf{u} \in \mathcal{T}_{\mathbf{x}}\mathcal{M}_{\mathcal{X}}$.

For an embedded manifold $\mathcal{M}_{\mathcal{X}}$ with intrinsic dimension $d$, we can assume the existence of an invertible global chart map $\xi : \mathcal{M}_{\mathcal{X}} \to \mathbb{R}^d$ mapping the manifold $\mathcal{M}_{\mathcal{X}}$ to its intrinsic coordinates. A vector $\mathbf{v}_{\mathbf{x}} \in \mathcal{T}_{\mathbf{x}}\mathcal{M}_{\mathcal{X}}$ on the tangent space of $\mathcal{M}_{\mathcal{X}}$ can be expressed as a pushforward $\mathbf{v}_{\mathbf{x}} = \mathbb{J}_{\xi^{-1}}(\mathbf{z})\mathbf{v}_{\mathbf{z}}$ of a tangent vector $\mathbf{v}_{\mathbf{z}} \in \mathbb{R}^d$ at $\mathbf{z} = \xi(\mathbf{x})$, where $\mathbb{J}$ indicates the Jacobian. Therefore, $\mathbb{J}_{\xi^{-1}}$ maps vectors $\mathbf{v}_{\mathbf{z}} \in \mathbb{R}^d$ into the tangent space of the embedded manifold $\mathcal{M}_{\mathcal{X}}$. The ambient metric $\mathrm{M}(\mathbf{x})$ can be related to the metric $\mathrm{M}(\mathbf{z})$ defined in terms of intrinsic coordinates via:

$$\mathrm{M}(\mathbf{z}) = \mathbb{J}_{\xi^{-1}}(\mathbf{z})^T \mathrm{M}(\xi^{-1}(\mathbf{z})) \mathbb{J}_{\xi^{-1}}(\mathbf{z}) . \tag{21}$$

In other words, we can use the metric $\mathrm{M}(\mathbf{z})$ to compute quantities on the manifold, such as geodesic paths. However, for an embedded manifold $\mathcal{M}_{\mathcal{X}}$, the chart map $\xi$ is usually not known. A workaround is to define the geometry of $\mathcal{M}_{\mathcal{X}}$ on another Riemannian manifold $\mathcal{M}_{\mathcal{Z}}$ with a trivial chart map $\xi(\mathbf{z}) = \mathbf{z}$ for $\mathbf{z} \in \mathcal{M}_{\mathcal{Z}}$, which can be mapped to $\mathcal{M}_{\mathcal{X}}$ via a smooth immersion $h$. In the next section, we elaborate on the connection between manifold learning and autoencoders.

### C.1. Deterministic AEs

We assume the decoder $h : \mathcal{Z} = \mathbb{R}^d \to \mathcal{X} = \mathbb{R}^G$ of a deterministic autoencoder is an immersion of a latent manifold with trivial chart map into a Riemannian manifold $\mathcal{M}_{\mathcal{X}}$ embedded in $\mathcal{X}$ and with metric $\mathrm{M}$. This is valid if one also assumes that $d$ is the intrinsic dimension of $\mathcal{M}_{\mathcal{X}}$. As explained before, the Jacobian of the decoder maps tangent vectors $\mathbf{v}_{\mathbf{z}} \in \mathcal{T}_{\mathbf{z}}\mathcal{M}_{\mathcal{Z}}$ to tangent vectors $\mathbf{v}_{\mathbf{x}=h(\mathbf{z})} \in \mathcal{T}_{\mathbf{x}}\mathcal{M}_{\mathcal{X}}$. The decoder induces a metric into the latent space following Equation (21) as

$$\mathrm{M}(\mathbf{z}) = \mathbb{J}_h(\mathbf{z})^T \mathrm{M}(h(\mathbf{z})) \mathbb{J}_h(\mathbf{z}) , \tag{22}$$

called *pullback metric*. The pullback metric defines the geometry of the latent manifold $\mathcal{M}_{\mathcal{Z}}$ compared to that of the manifold $\mathcal{M}_{\mathcal{X}}$. The metric tensor $\mathrm{M}(\mathbf{z})$ regulates the inner product of tangent vectors $\mathbf{u}_{\mathbf{z}}$ and $\mathbf{v}_{\mathbf{z}}$ on the tangent space $\mathcal{T}_{\mathbf{z}}\mathcal{M}_{\mathcal{Z}}$:

$$\langle \mathbf{u}_{\mathbf{z}}, \mathbf{v}_{\mathbf{z}} \rangle_{\mathcal{M}_{\mathcal{Z}}} = \mathbf{u}_{\mathbf{z}}^T \mathrm{M}(\mathbf{z}) \mathbf{v}_{\mathbf{z}} . \tag{23}$$

To enhance latent representation learning, distances in the latent space $\mathcal{Z}$ can be optimised according to quantities of interest in the observation space $\mathcal{X}$, following the geometry of $\mathcal{M}_{\mathcal{X}}$. For instance, we can define the length of a curve $\gamma : [0, 1] \to \mathcal{Z}$ in the latent space by measuring its length on the manifold $\mathcal{M}_{\mathcal{X}}$:

$$L(\gamma) = \int_0^1 \left\| \dot{h}(\gamma(t)) \right\| \mathrm{d}t$$

$$= \int_0^1 \sqrt{\dot{\gamma}(t)^T \mathrm{M}(\gamma(t))\dot{\gamma}(t)} \mathrm{d}t , \tag{24}$$

where the equality is derived by applying the chain rule of differentiation.

## C.2. Pulling Back the Information Geometry

In machine learning, exploring latent spaces is crucial, particularly in generative models such as VAEs. One challenge is defining meaningful distances in the latent space $\mathcal{Z}$, which often depends on the properties of stochastic decoders and their alignment with the observation space. Injecting the geometry of the decoded space of a VAE into the latent space requires a different theoretical framework, where the data is assumed to lie near a statistical manifold.

VAEs can model various data types by utilising the decoder function as a non-linear likelihood parameter estimation model. We consider the decoder's output space as a parameter space $\mathcal{H}$ for a probability density function. Depending on the data type, we express a likelihood function $p(\mathbf{x} \mid \boldsymbol{\varphi})$ with parameters $\boldsymbol{\varphi} \in \mathcal{H}$, reformulated as $p(\mathbf{x} \mid \mathbf{z})$ through a mapping $h : \mathcal{Z} \to \mathcal{H}$. We aim to define a natural distance measure in $\mathcal{Z}$ for infinitesimally close points $\mathbf{z}_1$ and $\mathbf{z}_2 = \mathbf{z}_1 + \delta\mathbf{z}$ when seen from $\mathcal{H}$. One can show that such a distance corresponds to the Kullback-Leibler (KL) divergence:

$$\text{dist}^2(\mathbf{z}_1, \mathbf{z}_2) = \text{KL}(p(\mathbf{x} \mid \mathbf{z}_1), p(\mathbf{x} \mid \mathbf{z}_2)). \tag{25}$$

To define the geometry of the statistical manifold, one can resort to information geometry, which studies probabilistic densities represented by parameters $\boldsymbol{\varphi} \in \mathcal{H}$. In this framework, $\mathcal{H}$ becomes a statistical manifold equipped with a FIM:

$$\text{M}(\boldsymbol{\varphi}) = \int_{\mathcal{X}} [\nabla_{\boldsymbol{\varphi}} \log p(\mathbf{x} \mid \boldsymbol{\varphi})][\nabla_{\boldsymbol{\varphi}} \log p(\mathbf{x} \mid \boldsymbol{\varphi})]^T p(\mathbf{x} \mid \boldsymbol{\varphi}) \, \mathrm{d}\mathbf{x}. \tag{26}$$

The FIM locally approximates the KL divergence. For a univariate density $p$, parameterised by $\varphi$, it is known that

$$\text{KL}(p(x \mid \varphi), p(x \mid \varphi + \delta\varphi)) \approx \frac{1}{2} \delta\varphi^{\top} \text{M}(\varphi) \delta\varphi + o(\delta\varphi^2). \tag{27}$$

In the VAE setting, we view the decoder not as a mapping to the observation space $\mathcal{X}$ but as a transformation to the parameter space $\mathcal{H}$. This perspective allows us to naturally incorporate the FIM into the latent space $\mathcal{Z}$. Consequently, the VAE's decoder can be seen as spanning a manifold $\mathcal{M}_{\mathcal{H}}$ in $\mathcal{H}$, with $\mathcal{M}_{\mathcal{Z}}$ inheriting the metric in Equation (26) via the Riemannian pullback. Based on this, we define a statistical manifold.

**Definition C.2.** A statistical manifold is represented by a parameter space $\mathcal{H}$ of a distribution $p(\mathbf{x} \mid \boldsymbol{\varphi})$ and is endowed with the FIM as the Riemannian metric.

The Riemannian pullback metric is derived as in Equation (22). Having defined the Riemannian pullback metric for VAEs with arbitrary likelihoods, one can extend the measurement of curve lengths in $\mathcal{Z}$ when mapped to $\mathcal{H}$ through $h$ as displayed by Equation (24). This approach allows flexibility in the choice of the decoder, as long as the FIM of the chosen distribution type is tractable.

# D. Learning Population Dynamics with Optimal Transport

The complexity of learning trajectories in high-dimensional data can be prevented by interpolating latent representations and decoding intermediate results to the data space for inspection. Here, we deal with learning population dynamics, which consists of modelling the temporal evolution of a dynamical system from unpaired samples of observations through time. As such, the task is naturally formulated as a distribution matching problem, and dynamic Optimal Transport (OT) has been a popular avenue for population dynamics.

Let the data be defined on a continuous space $\mathcal{X} = \mathbb{R}^d$. OT computes the most efficient mapping for transporting mass from one measure $\nu$ to another $\eta$, defined on $\mathcal{X}$. Relevant to dynamical systems, Benamou & Brenier (2000) introduced a *continuous formulation* of the OT problem. In this setting, let $p_t$ be a time-varying density over $\mathbb{R}^d$ constrained by $p_0 = \nu$ and $p_1 = \eta$. Dynamic OT learns a time-dependent marginal vector field $u : [0, 1] \times \mathbb{R}^d \to \mathbb{R}^d$, where $u_t(\mathbf{x}) = u(t, \mathbf{x})$. Such a field is associated with an Ordinary Differential Equation (ODE), $\mathrm{d}\mathbf{x} = u_t(\mathbf{x})\mathrm{d}t$, whose solution matches the source with the target distribution. Therefore, one can use dynamic OT to learn a system's dynamics from snapshots of data collected over time.

An efficient simulation-free formulation of dynamic OT comes from the OT Conditional Flow Matching (OT-CFM) model by Tong et al. (2024a) and Pooladian et al. (2023), who demonstrated that the time-resolved marginal vector field $u_t(\mathbf{x})$ has the same minimiser as the data-conditioned vector field $u_t(\mathbf{x}|\mathbf{x}_0, \mathbf{x}_1)$, where $(\mathbf{x}_0, \mathbf{x}_1) \sim \pi$ are tuples of points sampled

from the static OT coupling $\pi$ between source and target batches. Assuming Gaussian marginals $p_t$ and $\mathbf{x}_0$ and $\mathbf{x}_1$ to be connected by Gaussian flows, both $p_t(\mathbf{x}|\mathbf{x}_0, \mathbf{x}_1)$ and $u_t(\mathbf{x} \mid \mathbf{x}_0, \mathbf{x}_1)$ become tractable:

$$p_t(\mathbf{x} \mid \mathbf{x}_0, \mathbf{x}_1) = \mathcal{N}(t\mathbf{x}_1 + (1-t)\mathbf{x}_0, \sigma^2) \tag{28}$$

$$u_t(\mathbf{x} \mid \mathbf{x}_0, \mathbf{x}_1) = \mathbf{x}_1 - \mathbf{x}_0 \,, \tag{29}$$

where the value of $\sigma^2$ is a small pre-defined constant. Accordingly, the OT-CFM loss is

$$\mathcal{L}_{\text{OT-CFM}}(\xi) = \mathbb{E}_{t \sim \mathcal{U}(0,1), \pi(\mathbf{x}_0, \mathbf{x}_1), p_t(\mathbf{x}|\mathbf{x}_0,\mathbf{x}_1)} \left[ \|v_\xi(t, \mathbf{x}) - u_t(\mathbf{x}|\mathbf{x}_0, \mathbf{x}_1)\|^2 \right], \quad \text{with } t \sim \mathcal{U}(0,1). \tag{30}$$

Here, $v_\xi(t, \mathbf{x})$ is a neural network approximating the marginal vector field $u_t(\mathbf{x})$.

Given this formulation of dynamic OT, we highlight three aspects:

1. Dynamic OT only applies to continuous spaces.
2. OT-CFM benefits from low-dimensional representations since the OT-coupling is optimised from distances in the state space.
3. Based on Equation (28), OT-CFM uses straight lines to optimise the conditional vector field, thus *assuming Euclidean geometry*.

In the presence of discrete data like scRNA-seq counts modelled with VAEs, one can tackle (1) and (2) by learning dynamics in a low-dimensional representation of the state space, the latent space of a VAE with a discrete-likelihood decoder. Note, however, that (3) is still a shortcoming, since straight lines in the latent space of a VAE do not reflect geodesic paths on the decoded data manifold unless enforced otherwise, see Figure 1. In this work, we address this remaining limitation by regularising the VAE's latent space to induce locally Euclidean geometry, ensuring that straight lines in latent space better approximate geodesics on the decoded data manifold.

## E. Baseline Description

### E.1. GAE

#### E.1.1. THE MODEL

Here, we describe the Geodesic Autoencoder (GAE) from Huguet et al. (2022). For more details on the theoretical framework, we refer to the original publication. The GAE works by matching Euclidean distances between latent codes with the *diffusion geodesic distance*, which is an approximation of the diffusion ground distance in the observation space.

Briefly, the authors compute a graph with an affinity matrix based on distances between observations $i$ and $j$ using a Gaussian kernel as:

$$(\mathbf{K}_\epsilon)_{ij} = k_\epsilon(\mathbf{x}_i, \mathbf{x}_j) \,, \tag{31}$$

with scale parameter $\epsilon$, where $\mathbf{x}_i, \mathbf{x}_j \in \mathcal{X}$ and $\mathcal{X}$ is the observation space. The affinity is then density-normalised by $\mathbf{M}_\epsilon = \mathbf{Q}^{-1}\mathbf{K}_\epsilon\mathbf{Q}^{-1}$, where $\mathbf{Q}$ is a diagonal matrix such that $\mathbf{Q}_{ii} = \sum_j (\mathbf{K}_\epsilon)_{ij}$. To compute the diffusion geodesic distance, the authors additionally calculate the diffusion matrix $\mathbf{P}_\epsilon = \mathbf{D}^{-1}\mathbf{M}_\epsilon$, with $\mathbf{D}_{ii} = \sum_{j=1}^n (\mathbf{M}_\epsilon)_{ij}$ and stationary distribution $\boldsymbol{\pi}_i = \mathbf{D}_{ii}/\sum_j \mathbf{D}_{jj}$. The diffusion geodesic distance between observations $\mathbf{x}_i$ and $\mathbf{x}_j$ is

$$G_\alpha(\mathbf{x}_i, \mathbf{x}_j) = \sum_{k=0}^K 2^{-(K-k)\alpha} \|(\mathbf{P}_\epsilon)_{i:}^{2^k} - (\mathbf{P}_\epsilon)_{j:}^{2^k}\|_1 + 2^{-(K+1)/2}\|\boldsymbol{\pi}_i - \boldsymbol{\pi}_j\|_1 \,, \tag{32}$$

with $\alpha \in (0, 1/2)$. The running value of $k$ in Equation (32) defines the scales at which similarity between the random walks starting at $\mathbf{x}_i$ and $\mathbf{x}_j$ are computed.

Given the diffusion geodesic distance $G_\alpha$ defined in Equation (32), the GAE model is trained such that the pairwise Euclidean distances between latent codes approximate the diffusion geodesic distances in the observation space $\mathcal{X}$, in a batch of size $B$. Given an encoder $f_\psi : \mathbb{R}^G \to \mathbb{R}^d$, the reconstruction loss is optimised alongside a geodesic loss

$$\mathcal{L}_{\text{geodesic}}(\psi) = \frac{2}{B} \sum_{i=1}^N \sum_{j>i} (\|f_\psi(\mathbf{x}_i) - f_\psi(\mathbf{x}_j)\|_2 - G_\alpha(\mathbf{x}_i, \mathbf{x}_j))^2 \,. \tag{33}$$

### E.1.2. Additional comparison between FlatVI and GAE

Although related in scope, FlatVI significantly differs from the Geodesic Autoencoder (GAE) proposed by Huguet et al. (2022). Firstly, GAE is a deterministic autoencoder optimised for reconstruction based on a Mean Squared Error (MSE) loss. As such, the model is not tailored to simulate gene counts. On the contrary, FlatVI's decoder parameterises a Negative Binomial likelihood, allowing realistic generation of count data through sampling. This aspect has two advantages. By focusing on learning continuous parameters of a discrete likelihood, FlatVI explicitly models distributional properties of single-cell transcriptomics data, such as overdispersion, sparsity and discreteness. In contrast, a fully connected Gaussian decoder produces dense and continuous cells, failing to preserve the characteristics of the data. Moreover, the GAE regularises the latent space by approximating geodesic distances via a k-nearest-neighbour graph constructed in the observation space. This method requires the computation of pairwise Euclidean distances in the observation space. As suggested previously, gene expression is high-dimensional and, therefore, deceiving due to the curse of dimensionality. On the contrary, leveraging only the Jacobian and the output of the decoder to enforce latent space Euclideanicity, FlatVI is more suitable for larger datasets and eludes computing distances in high dimensions.

### E.2. Geometry-Aware Generative Autoencoder (GAGA)

### E.2.1. The model

GAGA (Sun et al., 2025) is an AE model trained such that latent distances approximate those on the data manifold, as estimated via PHATE (Moon et al., 2019). This estimation in data space enables the imposition of a pullback Riemannian metric in the latent space via the encoder, thereby aligning the geometries of the latent space and the data manifold. In addition, GAGA introduces a *warping approach*, wherein large distances are assigned to points outside the data manifold. The off-manifold status is determined using an auxiliary dimension in the latent space, which is trained adversarially.

The defined latent geometry supports various tasks, including (i) uniform sampling of the manifold, (ii) interpolation, and (iii) generation. In our experiments, we compare GAGA with FlatVI on task (ii). Specifically, on-manifold interpolations are achieved by training a neural ODE that minimises the curve length connecting two points on the manifold, based on the encoder's pullback metric. In other words, GAGA requires a parameterised neural network to define latent interpolations constrained to the manifold.

### E.2.2. Comparison between FlatVI and GAGA

GAGA and FlatVI share a common objective: To learn a latent geometry amenable to interpolation and manifold exploration. GAGA does so by learning distances on the manifold through a local neighbourhood approach, whereas FlatVI performs local manifold regularisation by assuming a metric defined on the statistical manifold spanned by the decoder of a discrete single-cell VAE. Consequently, FlatVI avoids the computation of pairwise distances for metric regularisation, relying on the assumption that aligning the pullback metric with an Euclidean geometry locally induces globally consistent manifold flattening, especially under dense sampling conditions.

In this context, FlatVI is explicitly designed for latent manifold regularisation in high-dimensional, over-dispersed count data, while GAGA assumes a continuous approximation of the gene expression space. Moreover, FlatVI encourages a simple and tractable latent geometry, whereas GAGA combines manifold regularisation with a parameterised neural network that approximates the shortest curve between pairs of points.

## F. Additional Notes on the Novelty of the Contribution

In VAE-based approaches, the decoder often parameterises a statistical manifold over the space of probability distributions. In our work, we explicitly model decoded single-cell profiles as points on the statistical manifold of negative NB distributions defined by the NB-VAE decoder. To the best of our knowledge, this is the first systematic approach to manifold learning in single-cell analysis that leverages the geometry of this specific statistical family.

This perspective is particularly impactful for scRNA-seq tasks that rely on interpolations in a latent or reduced space, such as trajectory inference and cellular fate mapping. Instead of interpolating in an arbitrary Euclidean latent space, we ask whether these transitions can be aligned with the intrinsic geometry of the NB statistical manifold. Since the NB distribution is widely accepted as the most accurate noise model for scRNA-seq data, enforcing such geometric consistency has the potential to produce more biologically meaningful trajectories.

# G. Complexity and Implementation

## G.1. Computational Complexity of the FIM Computation

We provide a breakdown of the computational complexity involved in evaluating the pullback FIM, as defined in Equation (20).

- We begin by analyzing the complexity of Equation (7), which generalizes Equation (20):

$$\mathrm{M}(\mathbf{z}) = \mathbb{J}_h(\mathbf{z})^\top \mathrm{M}(\boldsymbol{\varphi}) \mathbb{J}_h(\mathbf{z}) \,,$$

  where $\mathbb{J}_h(\mathbf{z})$ is the Jacobian of the decoder with respect to the latent variable $\mathbf{z}$.
- Let $G$ be the number of genes and $d$ the latent dimensionality, with $G \gg d$. The decoded mean parameter $\boldsymbol{\varphi} \in \mathbb{R}^G$ yields one value per gene, and the Jacobian $\mathbb{J}_h(\mathbf{z}) \in \mathbb{R}^{G \times d}$.
- The Fisher information matrix $\mathrm{M}(\boldsymbol{\varphi}) \in \mathbb{R}^{G \times G}$ encodes the second-order structure of the negative binomial likelihood with respect to the mean parameters.
- The product $\mathbb{J}_h(\mathbf{z})^\top \mathrm{M}(\boldsymbol{\varphi})$ costs $O(dG^2)$ and yields a $d \times G$ matrix. Multiplying this with $\mathbb{J}_h(\mathbf{z})$ gives a final cost of $O(d^2G)$. Since $G \gg d$, the total complexity is dominated by $O(dG^2)$.
- An equivalent result is obtained if one directly evaluates the sum of outer product matrices from Equation (20), as both forms represent the pullback metric.

Note that this analysis only reflects the cost of evaluating the pullback metric itself. In practice, computing $\mathbb{J}_h(\mathbf{z})$ requires evaluating the decoder $h$, which may be expensive.

## G.2. Implementation via the Jacobian-Vector Product

To compute the pullback metric efficiently, we use the Jacobian-vector product (JVP) instead of forming the full Jacobian. JVP is applied along each standard basis vector, dynamically assembling the necessary components while reducing memory and computational overhead. Our models, MLPs with one to three nonlinear layers, remain efficient despite the added cost. Notably, this approach scales better than GAE (Table 7), which requires pairwise distance computations for geodesic estimation.

# H. Model Setup

**Experimental details for Autoencoder models.** The Geodesic AE, NB-VAE, and FlatVI models are trained using shallow 2-layer neural networks with hidden dimensions `[256, 10]`. Batch normalisation is applied between layers, as we observed that it improves reconstruction loss. Non-linearities are introduced using the `ELU` activation function. All models are trained for 1000 epochs with early stopping based on the VAE loss and a patience value of 20 epochs. The default learning rate is set to `1e-3`. For VAE-based models, we linearly anneal the KL divergence weight from 0 to 1 over the course of training.

NB-VAE and FlatVI models use a batch size of 32, while Geodesic AE is trained with a batch size of 256, selected after evaluating $\{64, 100, 256\}$ based on validation loss. Notably, Geodesic AE is trained to reconstruct log-normalised counts, in contrast to NB-VAE and FlatVI, which model raw counts via a negative binomial decoder. To ensure training stability, all encoders receive inputs transformed via $\log(1 + \mathbf{x})$.

A summary of hyperparameter sweeps for FlatVI, along with selected values based on validation loss, is shown in Table 3.

*Table 3.* Hyperparameter sweeps for training FlatVI. The *Hidden dims* column excludes the final latent layer, which is fixed at dimension 10. Selected values used for main results are shown in **bold**.

|  | Batch size | Hidden dims | $\lambda$ |
|---|---|---|---|
| EB | **32**, 256, 512 | [1024, 512, 256], [512, 256], **[256]** | 0.001, 0.01, 0.1, **1**, 10 |
| Pancreas | **32**, 256, 512 | [1024, 512, 256], [512, 256], **[256]** | 0.001, 0.01, 0.1, **1**, 10 |
| MEF | **32**, 256, 512 | [1024, 512, 256], [512, 256], **[256]** | 0.001, 0.01, **0.1**, 1, 10 |

**Experimental details for OT-CFM.** To parameterise the velocity field in OT-CFM, we use a 3-layer MLP with 64 hidden units per layer and `SELU` activations. The learning rate is fixed at `1e-3`. The network receives a concatenation of the latent vector and a scalar time value as input, as required for conditional velocity estimation. Following recommendations from the official OT-CFM repository[1], we sample batches that include cells from all time points in each epoch. The variance hyperparameter $\sigma$ is set to 0.1 by default.

**The choice of the hyperparameter $\lambda$.** The hyperparameter $\lambda$ controls the strength of latent flatness regularisation in FlatVI (Table 1). While increasing $\lambda$ flattens the latent geometry (i.e., reduces curvature and variation of reconstruction), an overly large value can harm data fidelity and lower the model likelihood. A higher $\lambda$ leads to a more uniform (lower VoR) and flatter (lower CN) latent space.

We select $\lambda$ by evaluating the quality of trajectories inferred by OT-CFM on top of FlatVI embeddings. In practice, we increase $\lambda$ until trajectory reconstruction quality no longer improves. For most real datasets, increasing $\lambda$ from 0.1 to 1 enhances reconstruction performance. However, further increasing it to 10 offers no additional benefit. For the more complex MEF dataset, performance was best at $\lambda = 0.1$, so we retained that setting.

# I. Evaluation

## I.1. Metric Description

**Condition number.** Given a metric tensor $M(\mathbf{z})$, let $S_{\min}$ and $S_{\max}$ be its lowest and highest eigenvalues, respectively. The condition number (CN) is defined as the ratio

$$\mathrm{CN}(M(\mathbf{z})) = \frac{S_{\max}}{S_{\min}} \ . \tag{34}$$

Notably, an identity matrix has a CN equal to 1. The CN is an indicator of the stability of the metric tensor. A well-conditioned metric with a CN close to 1 suggests that the lengths and angles induced by the metric are stable. A large condition number means that the distances are more stretched in some directions than others. On an Euclidean manifold with a scaled diagonal metric tensor, distances are preserved in all directions.

**Variance of the Riemannian metric.** In assessing the Riemannian metric, we introduce a key evaluation called the Variance of the Riemannian Metric (VoR) (Pennec et al., 2006). VoR is defined as the mean square distance between the Riemannian metric $M(\mathbf{z})$ and its mean $\bar{M} = \mathbb{E}_{\mathbf{z} \sim p_{\mathbf{z}}}[M(\mathbf{z})]$. As suggested in Yonghyeon et al. (2021), we compute the VoR employing an affine-invariant Riemannian distance metric $d$, expressed as:

$$d^2(A, B) = \sum_{i=1}^{m} \left( \log S_i(B^{-1}A) \right)^2 \ , \tag{35}$$

where $S_i(B^{-1}A)$ indicates the $i^{th}$ eigenvalue of the matrix $B^{-1}A$. VoR provides insights into how much the Riemannian metric varies spatially across different $\mathbf{z}$ values. When VoR is close to zero, it indicates that the metric remains constant throughout. This evaluation procedure focuses solely on the spatial variability of the Riemannian metric and is an essential aspect of assessing the manifolds. Note that the expected value in Equation (35) is estimated using batches of latent observations with size 256.

**Velocity Consistency.** (Gayoso et al., 2024) This metric quantifies the average Pearson correlation between the velocity $v(\mathbf{x}_j)$ of a reference cell $\mathbf{x}_j$ and the velocities of its neighbouring cells within the k-nearest-neighbour graph. It is mathematically expressed as:

$$c_j = \frac{1}{k} \sum_{\mathbf{x} \in \mathcal{N}_k(\mathbf{x}_j)} \mathrm{corr}(v(\mathbf{x}_j), v(\mathbf{x})) \ . \tag{36}$$

Here, $c_j$ represents the velocity consistency, $k$ denotes the number of nearest neighbours considered in the k-nearest-neighbour graph, $\mathbf{x}_j$ is the reference cell, $\mathcal{N}_k(\mathbf{x}_j)$ represents the set of neighbouring cells. The value $\mathrm{corr}(v(\mathbf{x}_j), v(\mathbf{x}))$ is the Pearson correlation between the velocity of the reference cell $v(\mathbf{x}_j)$ and the velocity of each neighbouring cell $v(\mathbf{x})$. Higher values of $c_j$ indicate greater local consistency in velocity across the cell manifold.

---

[1] https://github.com/atong01/conditional-flow-matching

### I.2. Experiment Description

**Simulation details.** We simulate 10-dimensional negative binomial data from three distinct categories parameterised by means following distinct distributions and the same inverse dispersion. The negative binomial means $\boldsymbol{\mu}$ are drawn from 10-dimensional Gaussian distributions with category-specific means -1, 0 and 1. The inverse dispersion parameters $\boldsymbol{\theta}$ are again random and drawn from the same distribution across the different classes, namely a Gamma distribution with concentration equal to 2 and rate equal to 1. We exponentiate the means and take the absolute value of inverse dispersions to make them strictly positive. Note that we do not use size factors in the simulation experiment. Overall, we simulate 1000 observations drawn uniformly from different categories.

**Estimating latent pullback geodesics to generate Table 1 and Table 4.** We evaluate FlatVI trained with varying strengths of flattening regularisation, controlled via the hyperparameter $\lambda$. To assess the model's reconstruction performance, we consider how accurately FlatVI recovers the mean ($\boldsymbol{\mu}$) and inverse dispersion ($\boldsymbol{\theta}$) parameters used to simulate individual cells.

As an additional evaluation, we assess how well Euclidean distances in the latent space approximate geodesic distances on the latent manifold. To evaluate this similarity, we proceed in the following way:

- We sample pairs of simulated cells, encode them using FlatVI (under different $\lambda$ values) and obtain their latent representations $\mathbf{z}_1$ and $\mathbf{z}_2$.
- For each pair, we compute:
    1. The Euclidean distance between $\mathbf{z}_1$ and $\mathbf{z}_2$ in latent space.
    2. The pullback geodesic distance $d_{\text{latent}}(\mathbf{z}_1, \mathbf{z}_2)$ on the statistical manifold $\mathcal{M}_{\mathcal{Z}}$.

Geodesic distances are computed using the StochMan software[2], which approximates the shortest curve connecting two points on a manifold defined by a metric tensor. Specifically, we define the geodesic length via the KL divergence between infinitesimally close conditional distributions, leveraging the link between the Fisher information metric and the KL divergence of nearby decoded points (Equation (14), Equation (27)). The geodesic distance is thus formulated as:

$$d_{\text{latent}}(\mathbf{z}_1, \mathbf{z}_2) = \inf_{\gamma(t)} \int_0^1 \text{KL}\left(p(\mathbf{x} \mid h(\gamma(t))),\, p(\mathbf{x} \mid h(\gamma(t + \text{d}t)))\right)\, \text{d}t, \tag{37}$$
$$\text{where} \quad \gamma(0) = \mathbf{z}_1, \quad \gamma(1) = \mathbf{z}_2.$$

Here, $h(\gamma(t))$ denotes the decoder output at interpolated latent point $\gamma(t)$, and $p(\mathbf{x} \mid h(\gamma(t)))$ represents a negative binomial likelihood conditioned on those decoded parameters. In practice, we parameterise $\gamma(t)$ as a cubic spline $\hat{\gamma}(t)$ over 100 interpolation steps and optimise it to minimise the objective in Equation (37).

This procedure yields two vectors of pairwise distances (Euclidean and geodesic), which we compare in terms of their induced neighbourhoods and MSE.

**Spearman correlation between Euclidean and geodesic distances, and neighbourhood overlap metrics (Table 4).** We assess how closely the Euclidean and latent geodesic distances agree in terms of both correlation and local neighbourhood structure. Specifically, for VAEs trained with varying levels of regularisation, we perform the following procedure over 10 random repetitions:

- In each repetition, we randomly sample 50 simulated observations and encode them into the latent space.
- For all pairs of encoded points, we compute:
    1. The geodesic distances using the KL-based formulation in Equation (37).
    2. The Euclidean distances in the latent space.

Due to the computational cost of evaluating Equation (37), we limit the analysis to 50 observations per repetition. Once both pairwise distance matrices are obtained, we compute the following metrics:

- Spearman correlation: For each data point, we calculate the Spearman rank correlation between its Euclidean and geodesic distances to all other points and report the average across all data points and repetitions.
- Neighbourhood overlap: Using the Euclidean and geodesic distance matrices, we extract the $k$-nearest neighbours ($k = 3$ and $k = 5$) for each point and compute the average proportion of neighbours shared across both metrics. A

---

[2]https://github.com/MachineLearningLifeScience/stochman/tree/black-box-random-geometry

higher overlap indicates greater agreement between the local structures induced by Euclidean and pullback geodesic distances.

- Euclidean matrix distance: To quantify the overall similarity between the geodesic and Euclidean distances, we compute the MSE between the two full pairwise distance matrices. Lower values indicate better global agreement between the metrics.

**Riemannian metrics and path visualisation.** Riemannian metrics only take the metric tensor at individual latent codes as input (see Appendix I.1). The metric tensor for individual observations is calculated following Equation (11). The geodesic paths in Figure 2c are again approximated by cubic splines optimising the task in Equation (37). We compute such paths for 10 randomly drawn pairs in the region exhibiting high VoR in the unregularised VAE model for demonstration purposes.

**Trajectory reconstruction experiments.** In Table 2, we explore the performance of OT on different embeddings based on the reconstruction of held-out time points. For the EB dataset, we evaluate the leave-out performance on all intermediate time points. Conversely, on the MEF reprogramming dataset (Schiebinger et al., 2019) we conduct our evaluation holding out time points 2, 5, 10, 15, 20, 25 and 30 to limit the computational burden of the experiment. Note that the Pancreas dataset used for Figure 3 could not be used for this analysis, since it only has two time points: An initial and a terminal one.

After training OT-CFM excluding the hold-out time point $t$, we collect the latent representations of cells at $t-1$ and simulate their trajectory until time $t$, where we compare the generated cells with the ground truth via distribution matching metrics both in the latent and in the decoded space (mean $L^2$ and 2-Wasserstein distance in the latent space, 2- Wasserstein distance and MMD in the decoded space). Note that we replaced the $L^2$ distance with the MMD in the decoded space, as the former behaved uniformly across models in higher dimensions. For the latent reconstruction quantification, generated latent cell distributions at time $t$ are standardised with the mean and standard deviation of the latent codes of real cells at time $t$ to make the results comparable across embedding models. Results in Table 2 are averaged across three seeds.

**Fate mapping with CellRank.** We first train representation learning models on the Pancreas dataset. Then, following the setting proposed by Eyring et al. (2022), we learn a velocity field over the latent representations of cells by matching time points 14.5 to 15.5 with OT-CFM and input the velocities to CellRank (Lange et al., 2022; Weiler et al., 2024). Using the function `g.compute_macrostates(n_states, cluster_key)` of the GPPCA estimator for macrostate identification (Reuter et al., 2019), we look for 10 to 20 macrostates. If OT-CFM cannot find one of the 6 terminal states within 20 macrostates for a certain representation, we mark the terminal state as missed (see Figure 3). Terminal states are computed with the function `compute_terminal_states(method, n_states)`. Velocity consistency is estimated using the scVelo package (Bergen et al., 2020) through the function `scv.tl.velocity_confidence`. The value is then averaged across cells.

**Latent interpolation comparison with GAGA.** We train both GAGA and FlatVI on the pancreas dataset. For FlatVI, we use the same parameters as in Table 3. For GAGA, we choose hyperparameters to reflect the FlatVI setting, but make the neural network deeper, as it produced better results. Below we list the final array of hyperparameters used to train the GAGA model:

- `batch_size`: 256.
- `latent_space_dim`: 10.
- `activation_function`: ReLU.
- `learning_rate`: 0.001.
- `dist_reconstr_weights`: [0.9, 0.1, 0.0].

Once we optimise the models, we draw 100 pairs of multipotent cells and mature cells (a batch for each lineage). Pairing is done randomly. Then, we encode both multipotent and mature cells using both FlatVI and GAGA and interpolate the latent spaces of paired cells. For FlatVI, we interpolate linearly. For GAGA, we perform geodesic interpolations by training an energy-minimising neural ODE as explained in Sun et al. (2025) and Appendix E.2 over 200 epochs with a learning rate of 0.01. Intermediate results for both models are decoded with the respective decoder network, and the predicted expression is compared with the real gene expression distribution of lineage-specific markers.

# J. Data

## J.1. Data Description

**Embryoid Body (EB).** Moon et al. (2019) measured the expression of 18,203 differentiating human embryoid single cells across 5 time points. From an initial population of stem cells, approximately four lineages emerged, including Neural Crest, Mesoderm, Neuroectoderm and Endoderm cells. Here, we resort to a reduced feature space of 1241 highly variable genes. OT has been readily applied to the embryoid body datasets in multiple scenarios (Tong et al., 2020; 2024a), making it a solid benchmark for time-resolved single-cell trajectory inference. The data is split into 80% training and 20% test sets.

**Pancreatic Endocrinogenesis (Pancreas).** We consider 16,206 cells from Bastidas-Ponce et al. (2019) measured across 2 time points corresponding to embryonic days 14.5 and 15.5. In the dataset, multipotent cells differentiate branching into endocrine and non-endocrine lineages until reaching 6 terminal states. Challenges concerning such a dataset include bifurcation and unbalancedness of cell state distributions across time points (Eyring et al., 2022). The data is split into 80% training and 20% test sets.

**Reprogramming Dataset (MEF).** We consider the dataset introduced in Schiebinger et al. (2019), which studies the reprogramming of Mouse Embryonic Fibroblasts (MEF) into induced Pluripotent Stem Cells (iPSC). The dataset consists of 165,892 cells profiled across 39 time points and 7 cell states. For this dataset, we keep 1479 highly variable genes. Due to its number of cells, such a dataset is the most complicated to model among the considered. The data was split into 80% training and 20% test sets.

## J.2. Data Preprocessing

We use the Scanpy (Wolf et al., 2018) package for single-cell data preprocessing. The general pipeline involves normalisation via `sc.pp.normalize_total`, log-transformation via `sc.pp.log1p` and highly-variable gene selection using `sc.pp.highly_variable_genes`. 50-dimensional embeddings are then computed via PCA through `sc.pp.pca`. We then use the PCA representation to compute the 30-nearest-neighbour graphs around single observations and use them for learning 2D UMAP embeddings of the data. For the latter steps, we employ the Scanpy functions `sc.pp.neighbours` and `sc.tl.umap`. Raw counts are preserved in `adata.layers["X_counts"]` to train FlatVI.

## J.3. Details about Computational Resources

Our model is implemented in Python 3.10, and for deep learning models, we used PyTorch 2.0. For the implementation of neural-ODE-based simulations, we use the torchdyn package. Our experiments ran on different GPU servers with varying specifications: GPU: 16x Tesla V100 GPUs (32GB RAM per card) / GPU: 2x Tesla V100 GPUs (16GB RAM per card) / GPU: 8x A100-SXM4 GPUs (40GB RAM per card).

## K. Algorithms

**Train FlatVI.** Below, we provide a training algorithm for FlatVI.

---

**Algorithm 1** Train FlatVI

---

**Require:** Data matrix $\mathbf{X} \in \mathbb{N}_0^{N \times G}$, batch size $B$, maximum iterations $n_{\max}$, encoder $f_\psi$, decoder $h_\phi$, flatness loss scale $\lambda$
**Ensure:** Trained encoder $f_\psi$, decoder $h_\phi$, and inverse dispersion parameter $\boldsymbol{\theta}$
  Randomly initialize gene-wise inverse dispersion $\boldsymbol{\theta}$
  Randomly initialize the identity matrix scale $\alpha$ as a trainable parameter
  **for** $i = 1$ **to** $n_{\max}$ **do**
    Sample batch $\mathbf{X}^b \leftarrow \{\mathbf{x}_1, ..., \mathbf{x}_B\}$ from $\mathbf{X}$
    $\mathbf{l}^b \leftarrow \texttt{compute\_size\_factor}(\mathbf{X}^b)$
    $\mathbf{Z}^b \leftarrow f_\psi(1 + \log \mathbf{X}^b)$
    $\boldsymbol{\mu} \leftarrow h_\phi(\mathbf{Z}^b, \mathbf{l}^b)$
    $\mathcal{L}_{\text{KL}} \leftarrow \texttt{compute\_kl\_loss}(\mathbf{Z}^b)$
    $\mathcal{L}_{\text{recon}} \leftarrow \texttt{compute\_nb\_likelihood}(\mathbf{X}^b, \boldsymbol{\mu}, \boldsymbol{\theta})$
    $\text{M}(\mathbf{Z}^b) \leftarrow$ Equation (20)
    $\mathcal{L}_{\text{flat}} \leftarrow \texttt{MSE}(\text{M}(\mathbf{Z}^b), \alpha \mathbb{I}_d)$
    $\mathcal{L} = \mathcal{L}_{\text{recon}} + \mathcal{L}_{\text{KL}} + \lambda \mathcal{L}_{\text{flat}}$
    Update parameters via gradient descent
  **end for**

---

**Train OT-CFM on latent representations.** For time-resolved scRNA-seq data, cells are collected in $T$ unpaired distributions $\{\nu_t\}_{t=0}^T$. Individual time points correspond to separate snapshot datasets $\{\mathbf{X}_t\}_{t=0}^T$, each with $N_t$ observations. Every snapshot is mapped to tuples $\{(\mathbf{Z}_t, \mathbf{l}_t)\}_{t=0}^T$ of latent representations $\mathbf{Z}_t \in \mathbb{R}^{N_t \times d}$ and size factors $\mathbf{l}_t \in \mathbb{N}_0^{N_t}$ following the setting described in Section 3.1. We wish to learn the dynamics of the system through a parameterised function in the latent space $\mathcal{Z}$ of a VAE, taking advantage of its continuity and lower dimensionality properties.

---

**Algorithm 2** Train latent OT-CFM with FlatVI

---

**Require:** Datasets $\{\mathbf{X}_t\}_{t=0}^T$, variance $\sigma$, batch size $B$, initial velocity function $v_\xi$, maximum iterations $n_{\max}$, trained encoder $f_\psi$
**Ensure:** Trained velocity function $v_\xi$
  $\{\mathbf{Z}_t\}_{t=0}^T \leftarrow f_\psi(\{1 + \log \mathbf{X}_t\}_{t=0}^T)$
  $\{\mathbf{l}_t\}_{t=0}^T \leftarrow \texttt{compute\_log\_size\_factor}(\{\mathbf{X}_t\}_{t=0}^T)$
  $\{\mathbf{S}_t\}_{t=0}^T \leftarrow \texttt{timewise\_concatenate}(\{\mathbf{Z}_t\}_{t=0}^T, \{\mathbf{l}_t\}_{t=0}^T)$
  **for** $i = 1$ **to** $n_{\max}$ **do**
    Initialize empty array of velocity predictions $\mathbf{V}$
    Initialise empty array of velocity ground truth $\mathbf{U}$
    **for** $t_{\text{traj}} = 0$ **to** $T - 1$ **do**
      Randomly sample batches with $B$ observations $\mathbf{S}_{t_{\text{traj}}}^b, \mathbf{S}_{t_{\text{traj}}+1}^b$
      $\pi \leftarrow \text{OT}(\mathbf{S}_{t_{\text{traj}}}^b, \mathbf{S}_{t_{\text{traj}}+1}^b)$
      $(\mathbf{S}_{t_{\text{traj}}}^b, \mathbf{S}_{t_{\text{traj}}+1}^b) \sim \pi$
      $t \sim \mathcal{U}(0, 1)$
      $\mathbf{S}^b \leftarrow \mathcal{N}((1 - t)\mathbf{S}_{t_{\text{traj}}}^b + t\mathbf{S}_{t_{\text{traj}}+1}^b, \sigma^2 \mathbb{I}_d)$
      Append $v_\xi(t + t_{\text{traj}}, \mathbf{S}^b)$ to $\mathbf{V}$
      Append $(\mathbf{S}_{t_{\text{traj}}+1}^b - \mathbf{S}_{t_{\text{traj}}}^b)$ to $\mathbf{U}$
    **end for**
    $\mathcal{L}_{\text{OT-CFM}} \leftarrow \|\mathbf{V} - \mathbf{U}\|^2$
    Update parameters via gradient descent
  **end for**

---

**Size factor treatment.** Since the size factors $\mathbf{l}_t$, required for decoding, are observed variables derived from single-cell counts in the dataset, their values are not available when simulating novel cell trajectories from $t = 0$, hindering the use of the decoder to recover individual gene evolution. Assuming that the size factor is a real number and related to the cell state, we include $\log l_t$ in the latent dynamics and infer its trajectory together with the latent state representation $\mathbf{z}_t$. The log is taken for training stability. Therefore, we use OT-CFM to learn a velocity field $v_\xi : [0, 1] \times \mathbb{R}^{d+1} \to \mathbb{R}^{d+1}$ on the concatenated state $\mathbf{s}_t = [\mathbf{z}_t, \log l_t]$. The time-resolved vector field $v_\xi$ is modelled by matching subsequent pairs of cell distributions.

**Gene expression trajectories.** If the latent space $\mathcal{Z}$ can be mapped to the parameter manifold $\mathcal{H}$, trajectories in $\mathcal{Z}$ correspond to walks across the continuous parameter space via the stochastic decoder $h$. The temporal trajectory of the likelihood parameter vector $\boldsymbol{\mu}_t$ is given by

$$\boldsymbol{\mu}_t = h\left(\mathbf{s}_0 + \int_0^t v_\xi(t', \mathbf{s}_{t'})\, \mathrm{d}t'\right), \tag{38}$$

where $\boldsymbol{\mu}_0 = h(\mathbf{s}_0)$ and we express the decoder function $h(\mathbf{z}_t, l_t)$ from Equation (2) as $h(\mathbf{s}_t)$ for simplicity. Then, assuming a gene-wise constant inverse dispersion $\boldsymbol{\theta}_g$, discrete trajectories of gene counts follow the noise model $\mathbf{x}_t \sim \mathrm{NB}(\boldsymbol{\mu}_t, \boldsymbol{\theta})$.

# L. Additional Results

## L.1. Simulation Data

### L.1.1. SIMULATED DATA VISUALISATION

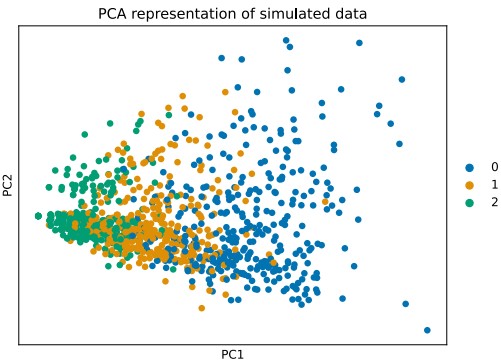

*Figure 6.* The PC dimensionality reduction plot of the simulated data coloured by category.

### L.1.2. COMPARISON WITH EUCLIDEAN SPACE METRICS

In Figure 7 we provide an extension to Figure 2 where we add how the VoR, CN and geodesic paths should appear in the Euclidean space. Notably, both CN and VoR are uniform, equating to 1 and 0, respectively. Furthermore, while some points in the simulated data exhibit high values for VoR and CN, the representation from FlatVI is more compatible with the expected one under Euclidean geometry than a normal NB-VAE. Additionally, path straightness is better preserved in FlatVI's latent geodesics compared to the counterpart, validating the purpose of our model.

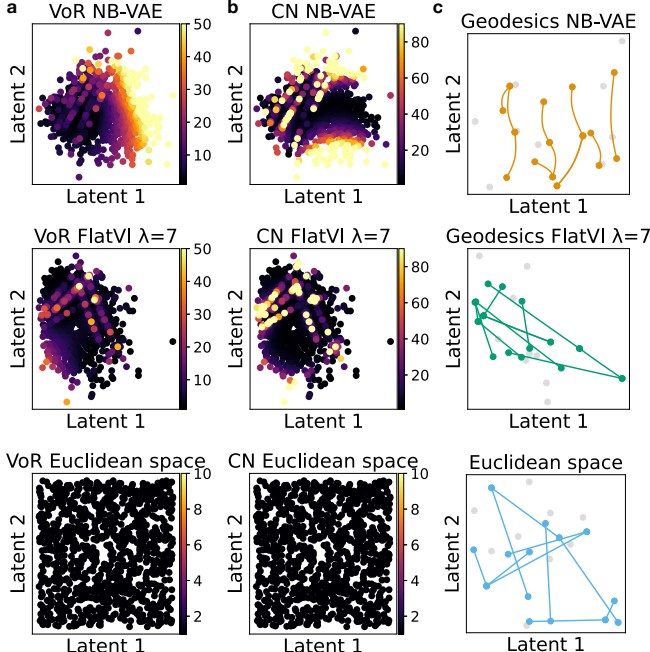

*Figure 7.* Comparison between the latent geometries of the NB-VAE (top row), FlatVI trained with $\lambda = 7$ (middle row) and an example Euclidean space of 1000 points evaluated in terms of **(a)** Variance of the Riemannian metric (VoR), **(b)** Condition Number (CN) and **(c)** straightness of the geodesic paths connecting pairs of latent points. The Euclidean panels are simulated as uniformly sampled points on a regular grid.

### L.1.3. ADDITIONAL METRICS

In addition to the metrics reported in Table 1, in Table 4 we provide further results that justify the usage of our flattening loss in the simulated data setting. The metrics are described in Appendix I.2. Such values represent how much the latent pullback geodesic and Euclidean distances and the neighbourhood structures deriving from them correspond. As can be inferred from the table, adding the regularisation promotes an overall improvement of the metrics. While Spearman correlation does not drastically separate results, adding the flattening regularisation marks an improvement in neighbourhood preservation metrics of up to 18% when considering 5 neighbours and 14% when using 3-point neighbourhoods. In other words, the pullback geodesics are better reflected by the Euclidean distance when applying our regularisation, which provides evidence of the working principles of our flattening loss.

*Table 4.* Comparison between FlatVI and the unregularised NB-VAE ($\lambda = 0$). Spearman (Geo-Euc) represents the Spearman correlation between latent Euclidean and pullback geodesic distances. Neighbourhood metrics measure the proportion of shared nearest neighbours (5-NN and 3-NN) across distance types. MSE (Geo-Euc) quantifies the absolute discrepancy between Euclidean and geodesic distances across 1000 pairs.

| $\lambda$ | Spearman (Geo-Euc) ($\uparrow$) | 5-NN overlap ($\uparrow$) | 3-NN overlap ($\uparrow$) | MSE (Geo-Euc) ($\downarrow$) |
|---|---|---|---|---|
| $\lambda = 0$ | $0.94_{\pm 0.00}$ | $0.50_{\pm 0.01}$ | $0.66_{\pm 0.00}$ | $47.75_{\pm 2.80}$ |
| $\lambda = 1$ | $0.95_{\pm 0.00}$ | $0.57_{\pm 0.01}$ | $0.63_{\pm 0.00}$ | $46.34_{\pm 6.45}$ |
| $\lambda = 3$ | $\mathbf{0.96}_{\pm 0.01}$ | $\mathbf{0.68}_{\pm 0.03}$ | $0.77_{\pm 0.00}$ | $16.74_{\pm 2.32}$ |
| $\lambda = 5$ | $\mathbf{0.97}_{\pm 0.01}$ | $0.58_{\pm 0.01}$ | $0.67_{\pm 0.01}$ | $8.65_{\pm 1.68}$ |
| $\lambda = 7$ | $\mathbf{0.97}_{\pm 0.01}$ | $0.60_{\pm 0.01}$ | $0.72_{\pm 0.01}$ | $\mathbf{8.02}_{\pm 0.67}$ |
| $\lambda = 10$ | $\mathbf{0.97}_{\pm 0.00}$ | $\mathbf{0.68}_{\pm 0.02}$ | $\mathbf{0.80}_{\pm 0.03}$ | $11.80_{\pm 1.04}$ |

### L.1.4. ANALYSIS OF SUB-OPTIMALLY FLATTENED REGIONS

In Figure 2, we show that introducing our flattening loss component in the VAE model training ensures a lower and more uniform Riemannian metric throughout the space, as well as lower CN. However, some points of our simulation dataset still display high decoding distortion and variance in the Riemannian metric as signals of insufficient flattening. In Figure 8, we show that latent paths between points of high VoR and their neighbours do display some curvature, indicating regions of the manifold with sub-optimal Euclideanisation. We investigated what causes points to exhibit a high VoR in both FlatVI ($\lambda$=7) and the regular NB-VAE. First, we found that regions of high VoR and CN in FlatVI tend to overlap, while in NB-VAE they are less correlated (see Figure 9a). Hence, while for most of the observations, flattening applies, the pullback metric from the decoder violates uniformity and preservation of angles and distances in some portion of the space.

We check the label annotation of the insufficiently flattened regions by overlaying their VoR and CN values onto the UMAP plot of the real data (see Figure 9b-c). By this analysis, we note that the high VoR and CN data points are concentrated at the inter-class boundaries in FlatVI's latent space (see Figure 10). In other words, observations in regions of the manifold enriched by different classes representing state transitions are more likely to fail to flatten. The fact that high VoR and CN are concentrated in regions of class heterogeneity may suggest that FlatVI fails to unfold some fast-changing manifold areas at the intersection between classes, and the decoder needs to violate the correspondence between the Euclidean latent space and the statistical manifold to ensure a proper reconstruction.

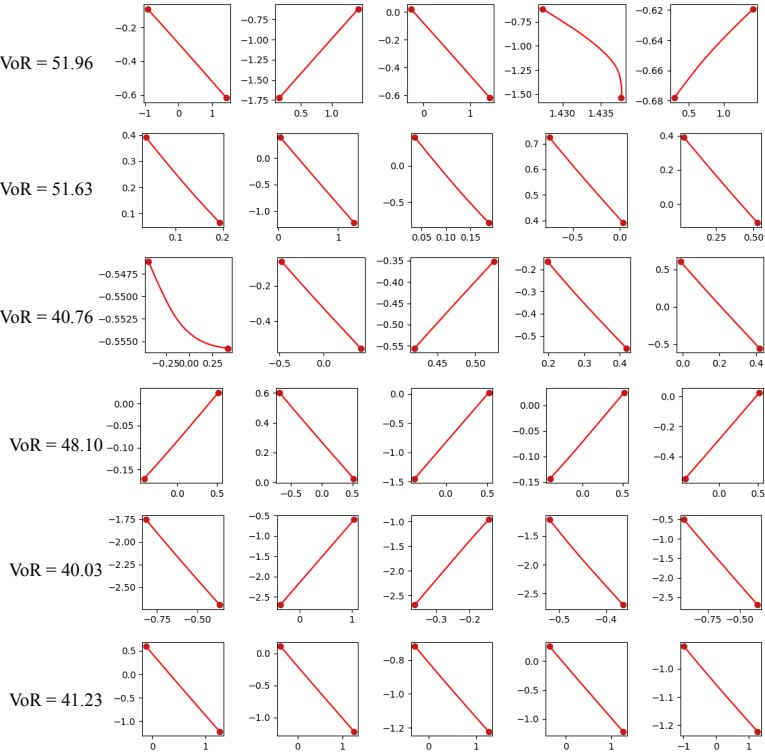

*Figure 8.* Geodesic paths between regions with high VoR and CN in the FlatVI embeddings with $\lambda = 7$. Every row represents a point with high VoR and CN. The columns are five randomly sampled neighbours in the dataset. Every plot represents the geodesic path between the point with high VoR and CN and the associated neighbour. As a representation of high VoR and CN, we select points with the VoR value larger than 30.

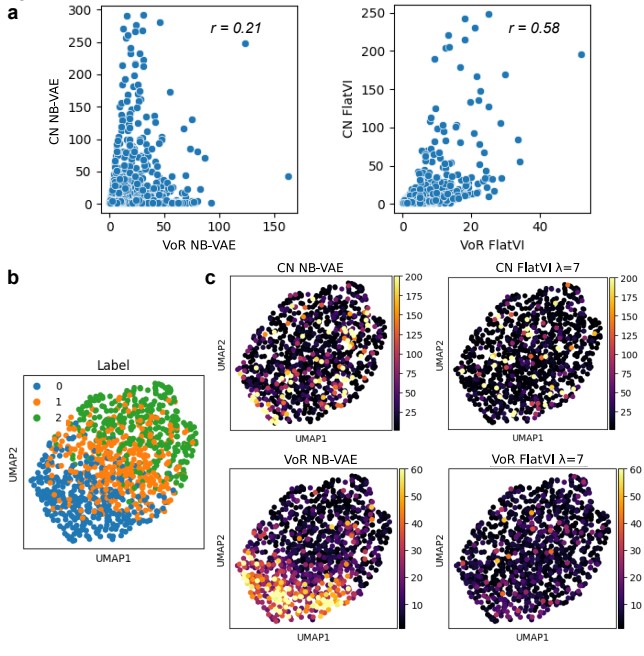

*Figure 9.* (**a**) The scatter plot and correlation values between VoR and CN in the embeddings computed by NB-VAE and FlatVI. (**b-c**) The UMAP plots of the real data coloured by class and CN and VoR values from the NB-VAE and FlatVI embeddings.

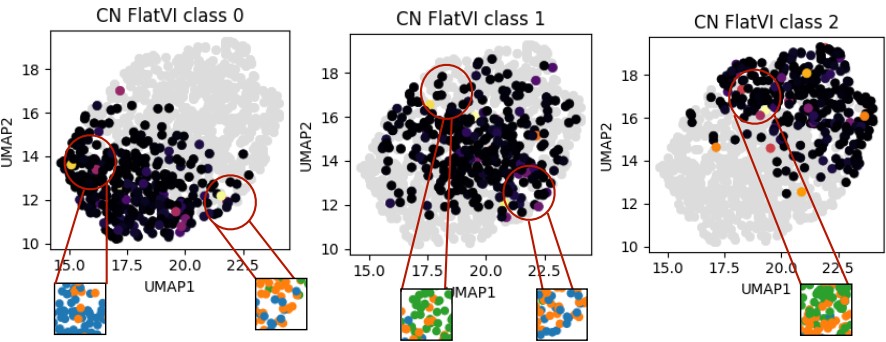

*Figure 10.* The UMAP plots computed on the simulated count data coloured by the CN from the FlatVI embeddings. Highlighted are regions with high VoR and CN. External boxes represent the label compositions of the highlighted regions (for a label-specific colour legend, see Figure 9b).

## L.2. Velocity Estimation

We compared our whole pipeline involving the combination of FlatVI and OT-CFM with the scVelo (Bergen et al., 2020) and veloVI (Gayoso et al., 2024) models for RNA velocity analysis. Figure 11 and Table 5 show examples of how our approach favourably compares with velocity estimation methods, namely inferring a proper velocity field in Acinar cells and detecting all terminal states with high velocity consistency in the Pancreas dataset.

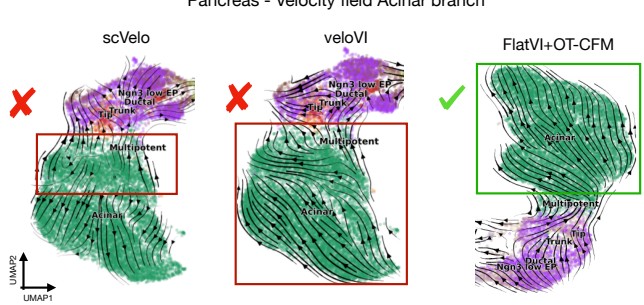

*Figure 11.* Comparison between the vector field learnt on the Acinar branch of the Pancreas dataset by using OT-CFM in combination with FlatVI and standard RNA velocity algorithms.

*Table 5.* Number of terminal states computed by CellRank and velocity consistency using the representations and velocities learnt by FlatVI+OT-CFM and standard RNA velocity algorithms.

| Method | Terminal states | Consistency |
|---|---|---|
| FlatVI + OT-CFM | **6** | **0.94** |
| veloVI | 5 | 0.92 |
| scVelo | 4 | 0.80 |

## L.3. Trajectory Visualisation of Real Data

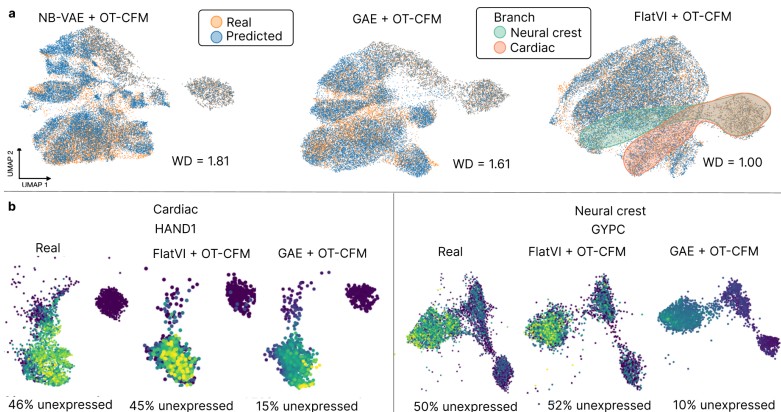

*Figure 12.* Prediction of scRNA-seq in time. (**a**) Overlap between real and simulated latent samples in the EB dataset. WD indicates the 2-Wasserstein distance between real and generated latent cell representations. (**b**) 2D UMAP plots of real and predicted cell counts from the cardiac and neural crest lineages of the EB dataset, comparing FlatVI and GAE as representations for OT-CFM. Colours indicate the predicted log gene expression of the reported lineage drivers GYPC and HAND1. Under each UMAP plot, we calculate the percentage of unexpressed marker instances along the trajectory.

## L.4. Additional Tables

*Table 6.* Univariate 2-Wasserstein distance between simulated and real marker gene expression for different lineage branches of the EB dataset across models. A lower value indicates that the model better approximates marker gene trajectories along the branch.

| | \multicolumn{3}{c}{2-Wasserstein real-simulated markers ($\downarrow$)} | | | | | | | | | |
|---|---|---|---|---|---|---|---|---|---|---|---|---|
| | Cardiac | | | Neural Crest | | | Endoderm | | | Neuronal | | |
| | GATA6 | HAND1 | TNN2 | NGFR | GYPC | PDGFRB | SOX17 | GATA3 | CDX2 | LMX1A | ISL1 | CXCR4 |
| GAE | 0.17 | 0.28 | 0.23 | 0.05 | 0.24 | 0.07 | 0.24 | 0.14 | 0.11 | 0.04 | 0.04 | 0.18 |
| NB-VAE | 0.03 | 0.24 | 0.07 | 0.07 | 0.09 | 0.04 | **0.08** | 0.07 | **0.02** | 0.02 | 0.06 | 0.07 |
| FlatVI | **0.02** | **0.09** | **0.03** | **0.02** | **0.05** | **0.03** | **0.08** | **0.02** | **0.02** | **0.01** | **0.02** | **0.03** |

*Table 7.* Runtime, in seconds, evaluated over a single forward pass considering different batch sizes for each compared model. The runtime is tested over 10 repetitions using random inputs with 2k dimensions.

| | \multicolumn{3}{c}{Runtime (s)} | | |
|---|---|---|---|
| Batch size | GAE | NB-VAE | FlatVI |
|---|---|---|---|
| 8 | $0.119_{\pm 0.012}$ | $0.000_{\pm .000}$ | $0.009_{\pm 0.004}$ |
| 16 | $0.095_{\pm 0.012}$ | $0.001_{\pm 0.000}$ | $0.007_{\pm 0.002}$ |
| 32 | $0.115_{\pm 0.012}$ | $0.001_{\pm 0.000}$ | $0.004_{\pm 0.003}$ |
| 64 | $0.099_{\pm 0.002}$ | $0.001_{\pm 0.000}$ | $0.005_{\pm 0.000}$ |
| 128 | $0.116_{\pm 0.007}$ | $0.002_{\pm 0.000}$ | $0.009_{\pm 0.004}$ |
| 256 | $0.218_{\pm 0.017}$ | $0.002_{\pm 0.000}$ | $0.009_{\pm 0.002}$ |
| 512 | $0.513_{\pm 0.021}$ | $0.002_{\pm 0.000}$ | $0.015_{\pm 0.004}$ |
| 1024 | $1.522_{\pm 0.013}$ | $0.003_{\pm 0.000}$ | $0.015_{\pm 0.001}$ |

*Table 8.* Separation between initial and terminal lineage states evaluated in terms of clustering metrics in the latent spaces of the distinct models. Different representation spaces are compared on how well they unroll developmental trajectories.

| | Silhouette Score ($\uparrow$) | | | Calinski-Harabasz ($\uparrow$) | | | Davies-Bouldin ($\downarrow$) | | |
|---|---|---|---|---|---|---|---|---|---|
| | EB | Pancreas | MEF | EB | Pancreas | MEF | EB | Pancreas | MEF |
| GAE | **0.28** | 0.15 | 0.09 | **1608.56** | 1723.13 | 11232.84 | **1.03** | 1.50 | 2.99 |
| NB-VAE | 0.19 | 0.26 | 0.21 | 940.87 | 2191.48 | 19440.38 | 1.28 | 1.56 | 2.35 |
| FlatVI | 0.21 | **0.50** | **0.31** | 983.41 | **6986.31** | **45372.75** | 1.18 | **0.73** | **1.50** |

## L.5. Additional comparison with GAGA

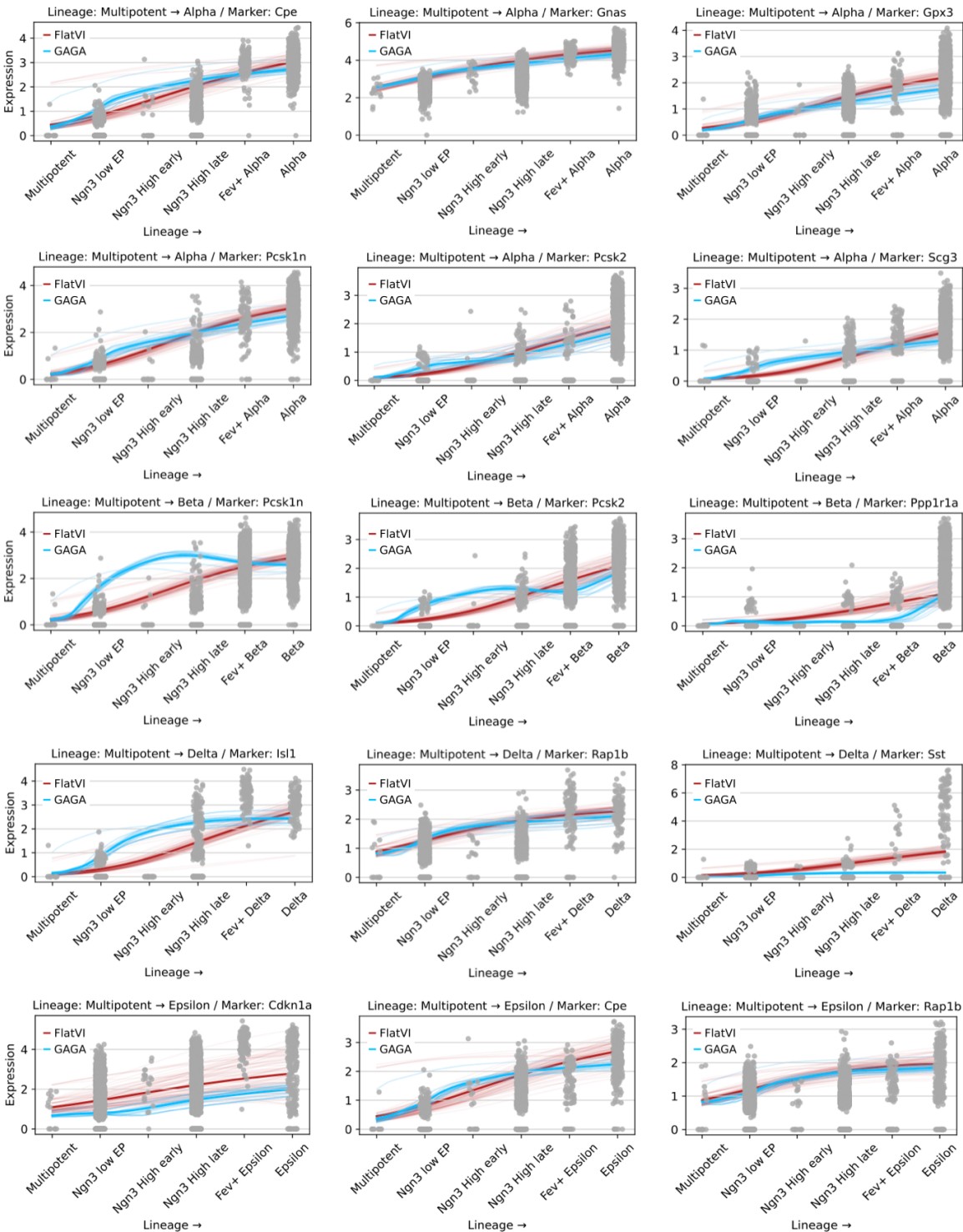

*Figure 13.* Decoded marker trajectories from latent interpolations on the pancreas dataset across multiple lineages, comparing FlatVI and GAGA. We draw 100 pairs of multipotent and mature cells from a terminal state, encode their gene expression and perform latent interpolation (linearly for FlatVI and with a neural ODE in GAGA). The intermediate states are decoded, and their marker gene expression is visualised together with the real marker expression distribution along the lineage (dark grey points). The individual trajectories are represented as low-opacity lines, while the centre solid line is the mean trajectory.

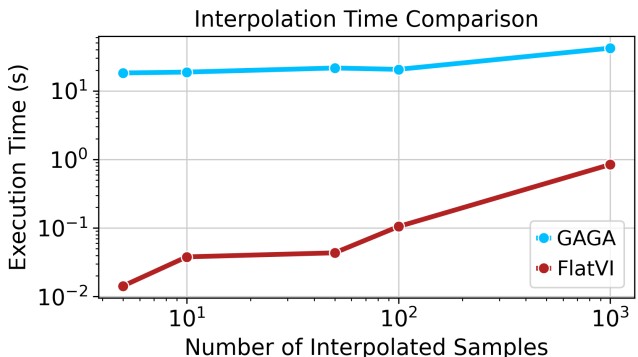

*Figure 14.* Scaling behaviour as a function of different numbers of interpolation samples for GAGA and FlatVI. On the y-axis, the execution time in seconds is presented on a log scale.

