# OpenReview forum: "Enforcing Latent Euclidean Geometry in Single-Cell VAEs for Manifold Interpolation"
_ICML.cc/2025/Conference — ICML 2025 spotlightposter_

### Official Review · Reviewer_zkLA · 2025-03-13

**Overall Recommendation:** 4

**Summary:**

This paper presents a novel training framework for VAEs for count data (i.e., negative binomial likelihood), where it enforces straight lines in latent space to map to geodesic paths on the data manifold, an assumption often made in VAE models for scRNA data but not explicitly enforced (or checked). In doing so, and combined with optimal transport-based modeling of cell fate transitions, FlatVI achieves good data reconstruction while better guiding the mapping between straight latent trajectories and cellular transitions.

## update after rebuttal
I thank the authors for their clarifications, and maintain my score.

**Claims And Evidence:**

The main claim is that, by adding the novel flattening loss to NB-VAE training, straight latent trajectories that is often used for interpolation would better correspond to minimal paths on the data manifold, e.g., by going through an intermediate cell state. As far as I can tell, the authors successfully demonstrate this on synthetic and multiple real datasets.

**Essential References Not Discussed:**

None I’m aware of.

**Experimental Designs Or Analyses:**

The experimental design and results seem comprehensive (checked all), and overall quite performant.

**Methods And Evaluation Criteria:**

The method is elegant and simple, adding just one additional loss term to achieve the desired result, though requiring an additional regularization hyperparameter (see Questions). The experiments and evaluation criteria chosen appear to be appropriate, and demonstrate FlatVI’s improved performance over existing baselines.

**Other Comments Or Suggestions:**

None

**Other Strengths And Weaknesses:**

- The paper is clearly and concisely written, and overall quite informative.
- The exposition from deterministic to stochastic AE was helpful to guide the reader.
- Really nice results overall and great visualization.

**Questions For Authors:**

- can you explain in more detail why alpha (in Eq 8) is a trainable parameter, and how varying a frozen alpha affects the results?
- could you clarify why the inverse dispersion parameter is not a function of the latent space (line 232-234), and whether it is estimated through the decoder (e.g., Eq. 2)? The evaluation metric considers reconstruction error on dispersion, so it seems that it is given by the decoder?
- What is the scale we should expect for the MSE between geodesic and euclidean distances? Is 0 the theoretical minimum if everything worked correctly?
- How should one choose lambda in practice, especially since the optimal for the three metrics do not necessarily coincide in Table 1 (though the differences in mu and theta are minimal)?
- Why are there these “streaks” of very high condition number for FlatVI in Fig 2b? While K.1.4 and Figure 7-9 touches on this point, I don’t quite see how the boundary effect highlighted in Fig 9 translate to the high CN regions in Figure 2?
- Naive question: in figure 4, imagining paths from initial to terminal state seems quite non-linear given the curved manifold (e.g., for Pancreas). Given that the trajectories should be straight and PCA is a linear reduction, wouldn’t we expect more linear manifolds and “straighter” paths from early to terminal? Instead of PCA and taking the first two PCs, is there a way to do dimensionality reduction but preserving the linear paths? Or am I fundamentally misunderstanding something?
- Another naive question, more broadly, why is it assumed (or desirable) that straight paths in latent space map to geodesic paths in data space through a cell’s transition?
- how would this scheme extend to work with, e.g., beta VAEs with an additional hyperparameter (on the KL loss) controlling the prior spread? It would be interesting to be able to regularize both the latent trajectory straightness and posterior variance?

**Relation To Broader Scientific Literature:**

This paper is a nice addition to the VAE, and more broadly, generative modeling literature. Having the correspondence between straight latent trajectories and geodesic trajectories on the data space would be useful in general for interpretability, and the specific negative binomial likelihood VAE would be useful in other applications with (low) count data, like neural recordings with spike counts.

**Theoretical Claims:**

I did not go through the loss proof in the amount of detail required to reconstruct it, but superficially everything checks out, aside from some questions in the end.

---

> ### Author Rebuttal · Authors · 2025-03-31
>
> We sincerely thank zkLA for their thorough review and positive assessment of our work, and we welcome the opportunity to address their remaining questions.
>
> > A1
>
> $\alpha$ relaxes the strictness of Euclidean regularisation. When the metric tensor $\mathrm{M}(\mathbf{z})$ is identity matrix $\mathbb{I}_d$, the latent space is strictly Euclidean, and the inner product reduces to the linear product. Our goal is to enforce a constant local geometry in the latent space. On a Riemannian manifold, distances are computed as in Eq.5 in the manuscript. If $\mathrm{M}(\mathbf{z}) = \mathbb{I}_d$ everywhere, the shortest path is a straight line. By setting $\mathrm{M}(\mathbf{z}) = \alpha \mathbb{I}_d$, we preserve straight geodesics while uniformly scaling tangent vectors. The trainable $\alpha$ provides flexibility in latent metric scaling, improving reconstruction; freezing it restricts optimal geometry learning to an assumed scale.
>
> > A2
>
> In single-cell analysis, $\theta$ is often modelled as a gene-specific parameter, capturing overdispersion from technical or biological effects. These are systematic gene-level patterns reflecting detection rates or expression variability. Thus, $\theta$ is learned as a free parameter by the likelihood model, not as a function of latent space. Gradients from the likelihood loss update both the VAE and gene-wise $\theta$. Though $\theta$ is independent of latent states, its estimation reflects reconstruction quality by interacting directly with the likelihood loss. Hence, we include it in Tab.1.
>
> > A3
>
> In Tab.1, we measure how closely straight paths align with geodesics by computing their MSE. Since geodesics on the latent manifold (equipped with the pullback metric) lack closed-form solutions without perfect regularisation, we approximate them numerically. Specifically, we use cubic splines that minimise the total Riemannian displacement between points, with local distances weighted by each model’s learned pullback metric (see l. 995-1023). When $\alpha=1$, the pullback metric reduces to the identity and the minimum MSE achievable is 0. In practice, our spline-based method introduces numerical errors, so the MSE rarely reaches 0. This metric remains robust for comparing how models deviate from ideal Euclidean behaviour.
>
> > A4
>
> Our simulations focus on understanding model behaviour and validating our claims. On real data, where selection matters, we recommend choosing $\lambda$ based on the downstream task. For trajectory inference, we tested $\lambda$ and selected, for each dataset, the highest possible $\lambda$ that improved trajectory reconstruction. For a detailed discussion, see App.G.
>
> > A5
>
> The mismatch occurs because Fig.9 displays a UMAP projection of the same latent encodings shown directly in Fig.2. While Fig. 2 presents the raw 2D latent space, we applied UMAP in Sec. K.1.4 to better visualize how unflattened regions are enriched in decision boundaries. Both figures represent identical data, the UMAP simply offers an alternative view for better visualisation. We will clarify this in the Appendix.
>
> > A6
>
> In Fig.4, we present data point embeddings rather than trajectories. We believe the non-linear structure observed, particularly for the Pancreas dataset, arises because PCA is a linear method that captures the global variance but does not preserve the local geometry of the data. Although our regularisation aims to approximate Euclidean space locally, some residual non-Euclidean structures may remain since we trained the model with $\lambda=1$. PCA, being a global method, emphasises this non-linearity when reducing the data to 2D. This is less apparent in the MEF dataset, where flattening appears to better unfold the natural temporal structure (Fig.4b). To better preserve the local structure, non-linear methods such as Isomap could be considered. We chose PCA for its interpretability and to analyse dimensionality across latent spaces.
>
> > A7
>
> Many single-cell tools assume that Euclidean distances in the representation space reflect biological relationships. FlatVI explicitly links the VAE's latent manifold to a locally Euclidean latent space, ensuring these distances better match biological proximity in terms of single-cell counts. This creates a principled approach for downstream analysis (kindly refer to the 1st answer to NJwg).
>
> > A8
>
> To extend FlatVI to $\beta-$VAEs with hyperparameters controlling the prior spread, we can incorporate the flattening loss alongside the modified KL term. This allows us to regularise both the latent trajectory straightness and posterior variance. Crucially, this extension does not disrupt the core guarantees of FlatVI, as the FlatVI loss only influences the latent space geometry shaped by the decoder model (Eq.7), while the KL term balances the posterior with the prior. This ensures that geometry regularisation and posterior variance are controlled without compromising the underlying structure of the latent space.

---

### Official Review · Reviewer_5Fan · 2025-03-14

**Overall Recommendation:** 4

**Summary:**

The manuscript focuses on learning a specific representation intended to understanding dynamics, specifically arising from single-cell RNA sequencing (scRNA-seq). The authors regularize the latent space of a Variational Autoencoder (VAE) using the geodesics of a statistical manifold. This approach ensures that straight lines in the latent space correspond to geodesics on the statistical manifold. The authors demonstrate the advantages of their method on toy datasets with known geodesics, as well as on real-world scRNA-seq datasets.

**Claims And Evidence:**

The claims are supported by theoretical and experimental results.

**Essential References Not Discussed:**

I believe there are no essential reference missing.

**Experimental Designs Or Analyses:**

I reviewed all the experiments and found them to be reasonable. However, I find Figure 2 difficult to interpret, and the conclusions drawn from the first two columns are unclear. However, I appreciate the quantitative experiments presented in Table 2.

**Methods And Evaluation Criteria:**

The method is theoretically grounded and is explained clearly. It has been rigorously tested on synthetic datasets with known geodesic properties, as well as on scRNA-seq datasets that exhibit temporal dynamics.

**Other Comments Or Suggestions:**

I don't have any other comments or suggestions.

**Other Strengths And Weaknesses:**

The manuscript is well written, and the method is clearly explained. I find most of the experiments to be of high quality; however, I find Figure 2 difficult to interpret.

I believe the experimental section could benefit from slight revisions. Specifically, I suggest moving some tables from the supplementary material, such as Table 4 and Table 8, into the main body of the text, as they present compelling qualitative results.

**Questions For Authors:**

I don't have any other questions.

**Relation To Broader Scientific Literature:**

The key contributions focus on enhancing the interpretability of the latent spaces in Variational Autoencoders (VAEs). Other studies have also attempted to regularize these latent spaces. The authors reference these related works and provide a comparison with one of them.

**Theoretical Claims:**

I checked the derivation of proposition 4.1 and it seems correct.

---

> ### Author Rebuttal · Authors · 2025-03-31
>
> We would like to thank Reviewer 5Fan for taking the time to review our work and for highlighting the aspects they found positive. We also appreciate their constructive suggestions regarding structural improvements and areas that could benefit from further clarification.
>
> > Interpretability of Fig. 2
>
> The first two columns of Fig. 2 illustrate how the values of the Condition Number (CN) and the Variance of the Riemannian Metric (VoR) are distributed across latent cell observations. We offer a brief, intuitive description of these metrics:
>
> * VoR: Measures how the Riemannian metric at a given latent point varies relative to its immediate neighbours. In a perfectly Euclidean latent space, this metric would remain constant throughout, as Euclidean manifolds exhibit uniform geometry. Therefore, we expect this metric to take low values in the regularised space.
>
> * CN: Quantifies how uniformly distances are scaled in different directions within the latent space geometry. Specifically, it represents the ratio between the largest and smallest eigenvalues of the metric tensor. In a perfectly Euclidean space, this ratio would be 1, meaning distances are preserved equally in all directions. A higher CN indicates greater distortion, with distances being stretched more in some directions than others.
>
> In the first two columns of Fig. 2, we show that the latent manifold produced by FlatVI consistently exhibits lower VoR and CN values compared to its unregularised counterpart, suggesting a closer approximation to Euclidean manifold properties. These results should be considered alongside Tab. 1, which demonstrates that stronger regularisation does not compromise reconstruction accuracy. In other words, FlatVI enforces the Euclidean constraint for downstream tasks while enabling precise trajectory reconstruction.
>
> Despite these improvements, some regions with elevated VoR and CN values remain in FlatVI’s output. We discuss these regions in Section K.1.4, where we hypothesise that they correspond to rapidly changing manifold areas located at the intersections between different cell-type categories.
>
> > Moving tables 4 and 8 into the main body.
>
> We agree with the reviewer and thank them for their suggestion. In case we get the chance to present a revised version with an extra page, we will add Tab.4 to Section 5.1 and Tab.8 to Section 5.4.

---

### Official Review · Reviewer_oZ7a · 2025-03-14

**Overall Recommendation:** 2

**Summary:**

The paper proposes FlatVI, a VAE training strategy that enforces local Euclidean geometry in the encoder’s latent space by regularizing the pullback metric of the decoder to approximate local Euclidean geometry in latent space. It validated the proposed method on simulated data and the single-cell trajectory inference task. It also demonstrated two other use cases of the locally Euclidean geometry in latent space, including latent vector fields and single-cell latent space visualization.

**Claims And Evidence:**

The authors make claims in the paper about straight paths and enforcing locally Euclidean geometry. This is only the case for globally euclidean data. Indeed curved manifolds can be locally euclidean and a lot of work has gone into showing how to drive geodesics there (see Neural FIM, Fasina et al. ICML 2023 ), (GAGA Xu et al. AISTATS 20250 , (Metric Flow Matching Kapusniak, Kacper, et al. 2024),.

**Essential References Not Discussed:**

This paper uses the pullback metric of the decoder to regularization the latent space but lacks citations on previous works in geometrically regularizing the latent space, such as Geometry-Aware Generative Autoencoder (Sun, Xingzhi, et al.) that regularizes the local latent space to match geodesic distances in data space, Geometry regularized autoencoders (
Duque, Andrés F., et al.), and Gromov-Monge Embedding (Lee et al., 2024), Neural FIM (Fasina et al ICML 20230.

**Experimental Designs Or Analyses:**

The paper showcased four experiments: 1) sanity check on simulated data, 2) single-cell trajectory inference, 3) latent vector field, and 4) single-cell representations.  The experiments showed that locally euclidean geometry is preserved in the latent space of FlatVI, but they do not lend strong support for why FlatVI is better for modeling single-cell data. Overall, the experiments are interesting but not compelling enough. Indeed they did not compare to Neural FIM, GAGA or metric flow matching.

The experiment on simulated data is more like a sanity check for whether the latent space is regularized as expected. The experiment on single-cell trajectory shows that FlatVI performs better than VAE without the flatness regularization and GAE. However, the interpolation method used in the experiment is OT Conditional Flow (OT-CFM), which assumes straight path interpolation, so the OT-CFM could compound the experiment results, making FlatVI more favorable. The paper can also benefit from benchmarking with more methods than just two methods.

The same can be said for the latent vector field experiment. Here, OT-CFM was used to compute the velocity fields, and then the vector fields were used to learn the terminal states of the Pancreas data. The experiment showed that FlatVI has more consistent velocity fields compared to the other two methods, but it’s not clear to me why consistent velocity is more ideal for single-cell data.

For the data representation, the paper claims that “FlatVI effectively represents the biological structure in the latent space as illustrated by the separation between initial and terminal states.”. However, in Figure 4, I do not see a clear advantage of FlatVI compared to the other methods, especially on EB and MEF.

**Methods And Evaluation Criteria:**

I think the idea of using the decoder’s pullback metric is very interesting, but I am unsure about the benefits of enforcing Euclidean geometry in latent space, especially for single-cell applications.

To my best knowledge, single cells are often hypothesized to lie on a manifold (manifold hypothesis), and the geometric and topological analysis of single-cell data have been proven very useful (Rizvi, A., Camara, P., Kandror, E. et al. Single-cell topological RNA-seq analysis reveals insights into cellular differentiation and development, Moon, K.R., van Dijk, D., Wang, Z. et al. Visualizing structure and transitions in high-dimensional biological data). Enforcing Euclidean structure might not be ideal for capturing the intrinsic non-Euclidean geometry often present in single-cell data.

The paper used OT Conditional Flow (OT-CFM) in their experiments, and it makes sense since OT-CFM does straight path interpolation, but there have been a lot of works recently leveraging the non-Euclidean geometry of the underlying manifold for trajectory inference tasks such as Metric Flow Matching (Kapusniak, Kacper, et al.), Geometry-Aware Generative Autoencoder (Sun, Xingzhi, et al.), WLF-SB (Neklyudov et al., 2024) etc.

**Other Comments Or Suggestions:**

Please put the contributions in relation to related work, and offer more insight on the global euclidean assumption.

**Other Strengths And Weaknesses:**

Overall, I think deriving a metric through the decoder pullback is very interesting, but the paper did not provide a compelling case for enforcing latent Euclidean geometry for single-cell data. The empirical experiments could benefit a lot from benchmarking with more interesting methods rather than the VAE baseline and GAE.

**Questions For Authors:**

Are there regularizations need to make the geometry globally euclidean ? For example can L_flat be modified?

**Relation To Broader Scientific Literature:**

The key contributions of this paper are about regularizing the latent space of VAE to approximate a locally Euclidean geometry through the pullback metric of the decoder. However, I am not convinced of the benefits of enforcing Euclidean geometry on the latent space, especially for single-cell data, considering there have been a lot of works showing the benefits of leveraging the non-Euclidean characteristics of the single-cell data (Rizvi, A., Camara, P., Kandror, E. et al. Single-cell topological RNA-seq analysis reveals insights into cellular differentiation and development, Moon, K.R., van Dijk, D., Wang, Z. et al. Visualizing structure and transitions in high-dimensional biological data) . They need to cite and discuss these as well as the other papers I mentioned above.

**Theoretical Claims:**

The paper derives the Fisher Information Metric for the negative binomial distribution. I scanned the proof and it seems reasonable..

---

> ### Author Rebuttal · Authors · 2025-03-31
>
> We sincerely thank oZ7a for their elaborate review. Their informed criticism offers us an opportunity to improve our submission. We will refer to two additional rebuttal figures stored at https://figshare.com/s/74ca822781c60b2a85f2.
>
> > Global regularisation
>
> Similar to GAE, we propose to:
> - Sample a couple of cells and encode them.
> - Approximate the latent geodesic distance between representations using the pullback. One could fit an energy-minimising cubic spline weighing local shifts by the pullback.
> - Minimise the difference between the latent Euclidean and the geodesic distances.
>
> This approach is expensive because it contains an unstable inner optimisation loop requiring decoding and gradient computation.
>
> > Local vs global
>
> While our flattening loss enforces local Euclidean geometry in the latent manifold, we emphasise that this regularisation is applied specifically in the regions where the data lies. Under the assumption that the latent manifold is sufficiently densely sampled, these local constraints can collectively approximate a globally Euclidean structure for the data-supported manifold. This assumption aligns with common practices in single-cell manifold estimation, where smooth transitions between neighbouring states define the manifold structure. We will add a clarifying assumption statement.
>
> > Contextualisation
>
> Apologies for the missing citations; we will add them upon revision.
>
> * We contextualize our model in App. E (ref. in l.239, col.2). Methodologically, FlatVI is a representation learning model, creating a space where tractable interpolations map to meaningful data-space paths. In contrast, MFM is an interpolation model, operating in a fixed PCA space and focusing on FM regularisation to align with a neighbourhood-based manifold. Unlike FlatVI, MFM does not account for probabilistic aspects of high-dimensional single-cell data.
>
> * More related are GAGA and NeuralFIM. Both methods perform latent geodesic regularisation to reflect neighbourhood structures in single-cell datasets, using k-NN graph-based approaches. Different from FlatVI, both models:
> 	* Only consider low-dimensional and continuous input data (e.g., PCA).
> 	* Do not enforce a tractable latent manifold, hence, they have to learn geodesics between latent observations with a NeuralODE.
>
> These aspects may hinder scaling to the gene space, as sparse expression vectors are unsuitable for Euclidean distances, and geodesic NeuralODEs are unstable (see below).
>
> > Manifold assumptions
>
> There is a distinction between the goals of the cited studies and our paper. For instance, PHATE produces a geometrically sound embedding of the data, exploiting neighbouring structures to **discover** patterns in the dataset. We implement a strategy that connects distances on a latent manifold to the geometry of high-dimensional and noisy discrete data to replace the standard VAE logic in combination with distance-based tools. FlatVI operates through the decoder function. When the decoder is expressive and assuming geodesic convexity, it reconstructs the data manifold's geometry and aligns our latent structure to the statistical manifold. Interpolating simpler proxy manifolds is established [1].
>
> > FlatVI's benefits
>
> Kindly refer to A1. to NJwg and A7. to zkLA.
>
> > Benchmark
>
> We focus on FlatVI’s representation aspect and how our guarantees improve downstream tasks. Our baselines, NB-VAE and GAE, reflect this. Comparing to NB-VAE serves as ablation to highlight our regularisation’s effects. GAE, like FlatVI, enforces Euclidean geometry via a k-NN-based approach, making the comparison fair.
>
> We validate our claims above by comparing FlatVI with GAGA (concurrent work under ICML guidelines). Both methods encode single-cell data into a latent space, where we interpolate between Multipotent and mature cells—using geodesic interpolations for GAGA and linear for FlatVI. Decoding the results, we assess marker gene reconstruction along lineages (Reb. Fig. 1a). FlatVI’s interpolations align better with true gene trends, whereas GAGA shows unnatural oscillations (Reb. Fig. 1a). GAGA’s simulations are also more unstable (Reb. Fig. 2a) and slower (Reb. Fig. 2b). Due to memory issues, we couldn’t run geodesic interpolations with NeuralFIM, but its training was slower than FlatVI (Reb. Fig. 2c).
>
> Since the FM algorithm is not the novelty of our work, we did not compare it with MFM.
>
> > Consistency
>
> Kindly refer to A2 to NJwg.
>
> > PCA - MEF and EB
>
> We agree that the sentence may sound ambiguous and overstated. In the text, we only mention the MEF and Pancreas datasets, where FlatVI better unrolls the dynamics. We quantified the separation between initial and terminal states in latent spaces using clustering metrics in Tab. 8. While our method outperforms competitors on MEF and Pancreas, the reviewer correctly noted that it does not show a clear advantage on EB. We will clarify this in the text.
>
> [1] de Kruiff et al., Pullback Flow Matching on Data Manifolds

---

### Official Review · Reviewer_NJwg · 2025-03-24

**Overall Recommendation:** 3

**Summary:**

The paper proposes FlatVI, a novel training framework for variational autoencoders (VAEs) applied to single-cell RNA-seq (scRNA-seq) data. Its central goal is to enforce Euclidean geometry in the latent space of discrete-likelihood VAEs—specifically, negative binomial VAEs commonly used for modeling gene expression counts—so that straight lines in the latent space correspond more closely to geodesics on the decoded statistical manifold.

The authors develop a flattening loss based on the pullback of the Fisher Information Metric (FIM) from the decoder, which they regularize toward a scaled identity matrix. The method is designed to retain data reconstruction performance while improving manifold geometry.

Empirical evaluations include:

- Simulations showing improved alignment between geodesic and Euclidean distances in latent space.
- Application with OT-CFM for scRNA-seq cellular trajectory reconstruction
- Application with CellRank for latent vector field and lineage mapping
- Some qualitative PCA-2D plots

## update after rebuttal

I appreciate the authors’ rebuttal, and the rebuttal Figure 1 serves as a new case study to show that the changes of three genes can be better estimated by FlatVI on the pancreas dataset. Therefore, I raised my score by 1 from broadline reject to broadline accept.

Nonetheless, I wish to clarify my stance (1) I agree this is a very interesting work extending VAE with a regularization to enhance euclidean metric (2) I think table 2 servers as a good benchmark to show the usefulness of this method on two real dataset (3) I am okay to use qualitative results in 5.4 to show the goodness of FlatVI representation at the end of main text to further enhance the work.

The main shortcoming from my perspective is the limited heuristic results for Section 5.3. Given that the authors only give one OT method as the basis for the analysis. Would the velocity consistency be enhanced at different levels of complexity of different datasets (maybe some datasets with some simple induced differentiation process and some complex organism developmental dataset like Mouse or C. elegans)? I feel like Figure 3 is a case study, but not comprehensive enough as the evaluation of a task. To this end, if the full section 5 consists of only one simulation, one task evaluation with 2 datasets, one case study, and one qualitative representation. I feel like the overall workload is under my expectations.

Moreover, I think rebuttal Figure 1 is a very nice addition to the results section. But again, to demonstrate the advantage of a method, we may consider a more systematic shows the usefulness among all the genes (or a subset of genes that are identified to be time-variant by a set of time analysis methods) in the dataset instead of curating three genes with better fitted trends.

In the end, I raised my score by one based on the rebuttal and the comments from zkLA and 5Fan. But I think ideally, Figure 3 and rebuttal Figure 1 should be expanded into two task evaluations to not only give a comprehensive heuristic evaluation but also better define this task to facilitate future work to build upon.

**Claims And Evidence:**

The paper makes three central claims:
1. Latent geometry regularization via flattening loss
2. Improved trajectory inference when combined with OT-CFM
3. Improved velocity field consistency in the latent space

I find the evidence for Claim 1 to be strong and clearly demonstrated through simulation experiments. The use of the pullback metric and flattening loss is well-motivated and effectively shown to induce Euclidean-like geometry in the latent space.

Claim 2 and Claim 3 are supported by results in Table 2 and Figure 3, respectively. However, I believe these claims highlight improvements that are largely specific to a narrow set of use cases. The improvements shown for trajectory inference rely on OT-CFM, which assumes Euclidean latent geometry. Many commonly used trajectory inference methods (e.g., pseudotime algorithms, diffusion-based or graph-based methods) operate in non-Euclidean or PCA-based spaces, where the benefits of FlatVI may not hold or may be less impactful. As such, the broader significance of the method is unclear when considering the wide variety of available alternatives in the single-cell analysis space.

For Claim 3, while Figure 3(b) reports improvements in “velocity consistency,” this metric reflects self-consistency of the learned vector field rather than ground-truth accuracy. It would strengthen the paper if the authors provided a clearer discussion about the biological interpretability and utility of velocity consistency in practical single-cell analyses.

Overall, I appreciate the authors’ effort and find the core idea to be well-presented. The work is technically sound and the writing is clear. That said, the methodological contribution is somewhat incremental, and the scope of demonstrated utility is limited to scenarios where specific assumptions (e.g., Euclidean latent geometry) hold. From my perspective, this work could be a useful enhancement to tools like scVI or CellRank, but may fall short of the bar for a standalone ICML paper in its current form.

**Essential References Not Discussed:**

.

**Experimental Designs Or Analyses:**

As discussed in Claims And Evidence.

**Methods And Evaluation Criteria:**

I am in general agree with the metric and criteria. Moreover, I think Table 2 is a good choice of benchmark showing OT-CFM got improved by FlatVI. It would be a good reference for later works to do trajectory reconstruction task.

**Other Comments Or Suggestions:**

.

**Other Strengths And Weaknesses:**

.

**Questions For Authors:**

- I would love to hear the authors thoughts about the significance of the FlatVI in trajectory inference task considering those alternative methods.
- I would love to the authors explain/discuss a bit more about their thoughts about the usefulness of velocity consistency in single cell analysis.

**Relation To Broader Scientific Literature:**

.

**Theoretical Claims:**

The proofs for theoretical claims look good to me.

---

> ### Author Rebuttal · Authors · 2025-03-31
>
> We thank NJwg for thoroughly reviewing our paper and providing constructive criticism to improve our submission. We kindly point the reviewer to the anonymous link https://figshare.com/s/74ca822781c60b2a85f2 containing the figures we refer to in some of our answers.
>
> > A1. Limited use/comparison with other algorithms.
>
> While our specific examples focus on OT-CFM, the benefits of FlatVI extend to a broader range of single-cell analysis techniques.
>
> * Many of the trajectory inference methods mentioned by the reviewer still fundamentally use distance calculations between representations of cells to define relationships and biological state proximity. For example, algorithms that build k-NN graphs (including pseudotime and diffusion) require a distance metric, and the Euclidean distance is often the default or a common choice.
> * By better aligning latent cell state Euclidean distances with biological distances in discrete gene expression counts (see our simulations), FlatVI's representation offers stronger guarantees for k-NN graph construction, which can be used by diffusion-based methods to assign cellular ordering and infer biological processes.
> *  In other words, FlatVI provides a representation space where biological distances are easy to compute and suitable for building temporal graphs and inferring cell ordering.
> *  The latent Euclidean assumption is also key in batch integration [1] and perturbation prediction [2], broadening the use of our regularisation.
> *  In Rebuttal Fig.1b and 2b-c (addressing oZ7a), we extend our comparisons with GAGA [5], a geodesic autoencoder, on the additional task of gene trajectory reconstruction across lineages. Here, we show that our simple Euclideanisation approach is faster, more stable, and more accurate than the most recent geodesic interpolation approach in capturing the dynamics of gene features on cellular manifolds.
>
>
> > A2. Velocity consistency
>
> Importantly, we evaluate our model with ground truth terminal state labels in Fig. 3a, showing that FlatVI’s representations guide Markov paths computed by CellRank to the correct terminal states, unlike competing methods. Consistency, introduced by VeloVI [3], measures vector field smoothness in a cellular representation space. Higher consistency means nearby latent profiles have correlated velocities, resulting in a smoother vector field. When estimating trajectories, related cellular states should have similar gradients, with gradual changes in vector field direction along the manifold. Regularising geometry before vector field estimation, as suggested in [3], enables smoother state transitions with Euclidean latent geometry.
>
> > A3. Contribution
>
> Our work presents a novel methodological formulation that, to the best of our knowledge, has not been explored before:
>
> - **Manifold learning:** Most manifold learning methods rely on continuous approximations of single-cell data. We are the first to define the cellular manifold as the negative binomial manifold spanned by the decoder of a discrete VAE. This means our geometric regularisation is directly informed by the statistical properties of single-cell data rather than assuming an arbitrary continuous structure.  Our simple approach leads to more biologically meaningful interpolations (Rebuttal figures).
> - **Pullback-based regularisation:** Unlike most existing methods that rely on k-NN neighbourhood estimation, we introduce pullback-based local geometric regularisation from the decoder of a VAE. This is a new approach to single-cell machine learning.
> - **Geometry-aware VAEs:** Our work extends existing geometry-aware VAEs by connecting latent Euclidean interpolations with geodesic paths on discrete statistical manifolds with tractable likelihoods. Similar geometric regularisation techniques have primarily been explored in deterministic autoencoders, but not in the probabilistic single-cell setting.
> - **Beyond trajectory inference:** While we focus on trajectory inference, the improved correspondence between latent Euclidean distances and statistical manifold geodesics has broader implications (see our first answer).
>
> Finally, we highlight that existing standalone ML conference papers in single-cell geometry [4, 5] often introduce new simulation-driven insights while targeting specific downstream tasks. FlatVI combines novel representation approaches with modelling assumptions in single-cell analysis.
>
> [1] Lücken et al., Benchmarking atlas-level data integration in single-cell genomics
>
> [2] Eyring et al., Unbalancedness in Neural Monge Maps Improves Unpaired Domain Translation
>
> [3] Gayoso et al., Deep generative modelling of transcriptional dynamics for RNA velocity analysis in single cells
>
> [4] Huguet et al., Manifold Interpolating Optimal-Transport Flows for Trajectory Inference
>
> [5] Sun et al., Geometry-Aware Generative Autoencoders for Warped Riemannian Metric Learning and Generative Modeling on Data Manifolds

---

### Decision · Program_Chairs · 2025-05-01

**Decision:**

Accept (spotlight poster)

**Comment:**

This is a methodologically novel and experimentally solid contribution at the intersection of machine learning, geometry, and computational biology. All reviewers appreciated the conceptual clarity and technical novelty, though some were hesitant due to limited comparative scope or concern over assumed geometry. However, the rebuttal mitigated these reservations, showing that FlatVI is not only a tractable regularizer but also a foundation for future geometric learning in probabilistic settings. The technique is widely applicable beyond scRNA-seq, such as in other count-based domains (e.g., spike train analysis, molecular measurements).